

# Sea-level rise along the Emilia-Romagna coast (Northern Italy) at 2100: scenarios and impacts

Luisa Perini[1], Lorenzo Calabrese[1], Paolo Luciani[1], Marco Olivieri[2], Gaia Galassi[3], and Giorgio Spada[3]

[1]Servizio Geologico, Sismico e dei Suoli, Regione Emilia-Romagna, Bologna, Italy
[2]Istituto Nazionale di Geofisica e Vulcanologia, Sezione di Bologna, Bologna, Italy
[3]Dipartimento di Scienze Pure e Applicate (DiSPeA), Università di Urbino, Urbino, Italy

*Correspondence to:* Giorgio Spada (giorgio.spada@gmail.com)

**Abstract.** As a consequence of climate change and human-induced land subsidence, coastal zones are directly impacted by sea-level rise. In some particular areas, the effects on the ecosystem and the urbanisation are particularly enhanced. We focus on the Emilia-Romagna coastal plain in Northern Italy, bounded by the Po river mouth to the north and by the Apennines to the south. The plain is ∼130 km long and is characterised by wide areas below sea level, in part reclaimed wetlands. In this context, several morphodynamic factors make the shore and back-shore unstable. During next decades, the combined effects of land subsidence and of the sea-level rise in consequence of climate change are expected to enhance the shoreline instability, leading to a further retreat. The consequent loss of beaches would impact the economy of the region, tightly connected with tourism infrastructures. Furthermore, the loss of wetlands and dunes would threaten the ecosystem, crucial for the preservation of life and environment. These specific conditions show the importance of a precise definition of the possible local impacts of the ongoing and future climate variations. The aim of this work is the characterisation of vulnerability in different sectors of the coastal plain and the recognition of the areas in which human intervention is urgently required. The IPCC AR5 sea-level scenarios are merged with new high resolution terrain models, current data for local subsidence and predictions of a flooding model (`in_CoastFlood`) to develop different scenarios for the impact of sea-level rise to year 2100. First, the potential land loss due to the combined effect of subsidence and sea-level rise is extrapolated. Second, the increase of floodable areas in consequence of storm surges is quantitatively determined. The results are expected to support the regional mitigation and adaptation strategies designed in response to climate change.

## 1 Introduction

The sea-level rise associated with global warming is a growing concern for the scientific community, as well as for governments, the media and the public. Climate-driven sea-level rise has a direct impact on the coastal zones, where it has threatening consequences on the ecosystem and the urbanisation. The causes of contemporary sea-level rise have been reviewed by Nicholls and Cazenave (2010), who have also identified, globally, the coastal areas that are particularly vulnerable to flooding. These include low-elevation coastal zones, densely populated areas where the natural or human-induced rate of subsidence is appreciable, and regions characterised by a limited adaptation capacity. In addition, higher sea levels are expected to amplify flooding caused by storm surges and hurricanes, with an enhanced impact on population (Nicholls et al., 1999). The last Intergovernmental Panel




on Climate Change (IPCC) Fifth Assessment Report (AR5)  has made a considerable progress toward an understanding of the 20th century sea-level rise and its variability on a global scale; furthermore sea-level projections for the 21th century have been greatly improved (Church et al., 2013). Nevertheless, detailed projections of sea-level rise at regional and local scale are still hampered by the complex local response of the coastal system to sea-level rise (Cazenave and Cozannet, 2014; Hinkel et al.,

5    2015).

In the general context outlined above, this work focuses on the possible effects of sea-level rise on the coastal area of Emilia-Romagna (Northern Italy), one of the administrative regions of Italy facing the Northern Adriatic Sea (see Figure 1).    F1 The Emilia-Romagna (E-R) coastal plain has a crucial economical and naturalistic value. It is, indeed, the site of one of the largest tourism industries in Europe, the *Riviera Romagnola*, and it embraces the Po Delta protected area, which was recently

included in the Man and the Biosphere Programme by Unesco[1]. Characterised by a gentle slope, the E-R coastal plain is high vulnerable in consequence of *i)* widespread coastal erosion, *ii)* seawater intrusions related to sea-level rise, *iii)* storm surges and *iv)* land subsidence. These four major threats have been exacerbated by the human pressure that took place since the 1940s, when a rapid urbanisation and a strong exploitation of underground resources (water and gas) initiated (Lorito et al., 2010; Perini and Calabrese, 2010). The reader is referred to the recent work of Aucelli et al. (2016) for the risk assessment of coastal

inundation in another Italian area characterised by spread back-shore depressions and subsidence (the Volturno coastal plain in southern Italy). A nationwide, detailed account for sea-level rise and potential drowning of the Italian coastal plains has been given by Antonioli et al. (2017). A more sophisticated approach was taken by Wolff et al. (2016) who downscaled the Dynamic Interactive Vulnerability Assessment (DIVA) model (Hinkel et al., 2014) to the E-R contest. This model considers the effects of sea-level rise in terms of impacts on the population and on the existing assets. It also foresees that mitigation actions, as the

building of dykes, will gradually accompany the sea-level rise to year 2100.

The awareness by the local authority (the Emilia-Romagna Regional Administration), that a holistic approach was desirable to properly manage the climate-related risks at coastal areas, motivated the set up of an integrated path, called "Gestione Integrata delle Zone Costiere" (GIZC) which led to the approval of specific guidelines[2]. The document recognises the need of putting the mitigation of local risk in the climate change framework, with a special attention to the sea-level rise issue.

Subsequently, in 2007, the EU delivered the so-called "Floods Directive"[3] (FD), aimed to reduce and manage the risks posed by floods pose to human health, the environment, cultural heritage and economic activity. The recent Italian Environmental Ministry guidelines on coastal protection from erosion and climate change effects[4], delivered in cooperation with coastal regions in 2016 and in accordance with the FD, also highlight recommendations of the Emilia Romagna GIZC guidelines. The document remarks that long term climate changes could not be excluded from the national and regional plans for the mitigation

of the seawater flooding risks. These concepts are also included in the nationwide SNAC (National Strategy for the Adaptation to Climate Change)[5]. The development of these national and transnational laws and guidelines drove the definition, for the

---

[1] See goo.gl/gi1yNE

[2] "*Linee Guida GIZC*", deliberation of Regional Council n. 654/2005.

[3] Directive 2007/60/EC on the assessment and management of flood risks (goo.gl/pzJ74o).

[4] *Linee Guida Nazionali per la difesa della costa dai fenomeni di erosione e dagli effetti dei cambiamenti climatici* (goo.gl/QAL8iN)

[5] *Strategia Nazionale di Adattamento ai Cambiamenti Climatici,* 2014 (goo.gl/zCYV6C)



Emilia-Romagna Regional Administration, of the "Path toward a strategy for the mitigation of climate changes"[6]. This path includes the study of the impact on climate change on coastal zones and the identification of the most vulnerable areas by means of models and cartographic analyses.

   Within the multidisciplinary pathway above, it is crucial to identify and characterise the most vulnerable portions of the
coastal plain, on which prevention and mitigation actions should be focused at first. Local and short term phenomena occurring in the E-R coastal plain and in its neighbourhoods have been put in a long term framework, which also includes regional and global climate change. The attention on the combined effects of climate-induced sea-level rise, of subsidence and of meteo-marine events on the E-R coast arouse two decades ago, following the 1992 IPCC Supplementary Report (Intergovernmental Panel on Climate Change, 1992). Bondesan et al. (1995) preliminarily investigated the impact of these IPCC scenarios for
sea-level rise to year 2100, in combination with expected subsidence and trends for storm surges. They concluded that the area potentially affected by the risk of flooding was doomed to increase in the future. For the first time, in the work of Gonella et al. (1998) a numerical model based upon geographical information system (GIS) tools was applied to the 1992 IPCC scenarios, considering storm surge return periods of 1, 10, and 100 years. The analysis was also extended to the risk assessment in terms of saltwater ingression and soil consumption. Recently, Antonioli et al. (2017) have used IPCC AR5 global projections at 2100
for sea-level rise with local assessment for subsidence derived from GPS and TG data. These have provided regional scenarios for different parts of the low elevation portion of the Italian peninsula, including the E-R coastal plain.

   The main purpose of this work is the determination of the future impact of land subsidence and sea-level rise on the E-R coast. In particular, for the first time in this region, the focus will be on two aspects: *i)* the loss of territories that are currently at, or above, sea level (Case study #1, CS1) and *ii)* the enlargement of the areas potentially affected by seawater flooding (Case
study #2, CS2). Our objective is twofold: *i)* the identification of those portions of the E-R coast which potentially could be flooded by the end of current century in the occasion of storm surges, and *ii)* the comparison of the results with the current hazard map for the ingression of the sea that resulted from the "Flood Directive", hereafter referred to as FD (European Parliament and Council, 2007). In the following, CS1 and CS2 will be discussed taking the geomorphological and geodynamic complexity of the area into account, and adopting the sea-level projections published in the last IPCC Fifth Assessment Report
(AR5) (Church et al., 2013).

   The paper is organised as follows. Section 2 describes the physical characteristic of the study region and the driving processes. The data and methods employed are presented in Section 3, followed by the results in Section 4. Our conclusions are drawn in Section 5.

## 2  Physical characteristics and driving processes

The E-R coastal plain develops along the south-eastern margin of Po Valley in Northern Italy; it is bounded to the north by the Po di Goro branch of the Po Delta and to the south by the Apennine Chain. The orientation of the chain and the direction of progradation of the Po River make the plain narrow to the south (a few kilometres in the EW direction), and broad to the north

---

[6]Deliberation n. 2200/2015, *Percorso verso una strategia di mitigazione dei cambiamenti climatici.*



where it exceeds $\sim$40 km (see Figure 2). The shoreline is about 130 km long and is characterised by low-elevation sandy beach **F2**
ridges, occasionally associated with lagoons, wetlands and river mouths.

The coastal area and its shoreline can be divided into a northern (N) and a southern (S) sector, whose boundary can be placed around the town of Cervia (see Figure 1). The different physical characteristics of the two sectors, which are of relevance for

the determination of the response of the coastal area to the marine ingressions, are detailed in Perini and Calabrese (2010). The vulnerability of the entire coastal area is mainly caused by the absence of dunes and by their discontinuity, especially in the S sector. A further cause of vulnerability is the presence, particularly in the N sector, of wide areas that are currently placed below sea level. Moreover, the combination of subsidence with the natural retreat of the shoreline due to the reduction of the river sediment discharge is responsible for a widespread coastal erosion, both on short and long term time scales (Perini and

Calabrese, 2010).

The human pressure and the underground exploitation of the E-R coastal plain and of its surroundings have enhanced the expected effects of climate change, and particularly of sea storms. The risk for sea flood has been magnified by the urbanisation that, since the end of the second World War, was increased by $\sim$400% in terms of occupied area (Lorito et al., 2010). In front of most coastal towns and resorts the dunes, which are the natural rampart against the sea flood and a natural reservoir of sand

for the nourishment of the beaches, have been totally destroyed by urbanisation. In some cases the shorelines retreat sometimes have reached the buildings and infrastructures. This urbanisation, besides, is mostly concentrated along a narrow area along E-R coastal plain, bordering on the zone where normally most of the energy is dissipated during intense meteo-marine phenomena.

The Emilia-Romagna shoreline is highly artificialized and shows different types of coastal flood defences. These are mainly composed by emerging or submerged longitudinal breakwaters and localised jetties, groynes and seawalls, especially in the

Goro lagoon. In the N sector, following the extreme flooding event that occurred in 1966 (Perini et al., 2011), a long, 4 m high embankment was erected to protect the back-shore, at a distance ranging between 0.5 and 1.5 km from the shoreline. However, during the 1970s and 1980s, the vulnerability of the area between the shoreline and the wall was not considered a concern and an intensive urbanisation took place there. In the S sector, on the other hand, no coastal defences has been erected against marine inundations, while the sparse flood protection dykes are only aimed to defend local infrastructures from sea storms.

Against the sea storms, it is common practice off-season (*i.e.,* during the winter) to build temporary embankments along the beaches. These, if properly designed, play a crucial role in the attenuation and mitigation of the effect of sea storms, in reducing the ingression of sea water in the back-shore and in preventing erosion.

For each of the two sectors considered (N and S), the physical features of the E-R coastal plain are summarised in Table 1. **T1** These include the main geometrical features (the coastal plain width $D$ and its elevation above sea level $H$), geomorphological

characteristics, shapes and tendency, shoreline width $D_S$ and average width $\bar{D}_S$, and shoreline backside usage. Furthermore, in this Section, we provide an account of the processes that are responsible for the coastal vulnerability of the area. These are: *i)* land subsidence, *ii)* sea-level variability, and *iii)* storm surges.





## 2.1 Land subsidence

For those areas slightly above sea level, the main consequence of subsidence is an increased vulnerability to the effects of storm surges, since the progressive lowering of the ground below sea level will facilitate the marine ingression. Current average rates of land subsidence observed in the S sector of the E-R coastal plain, as well as in the southernmost part of the N sector, are close to 5 mm yr$^{-1}$. However, in some areas of the S sector and particularly at the mouths of rivers and in the back-shore of the Cesenatico and Rimini areas, they may exceed ∼10 mm yr$^{-1}$. In the N sector, rates of subsidence are smaller, typically in the range of 0 to 2.5 mm yr$^{-1}$, while in the area of the Po Delta rates of ∼10 mm yr$^{-1}$ are again observed (Arpa-RER, 2012) (see Figure 3). Locally, land subsidence reaches 20 mm yr$^{-1}$ due to gas extraction. This occurs, in particular, at the Fiumi Uniti river mouth. These significant present-day rates can result in a total subsidence of 2-3 meters in a century (Bondesan et al., 1995), as a consequence of the combined action of natural and anthropogenic factors.

The natural component of land subsidence is normally the result of tectonic activity, Glacial Isostatic Adjustment (GIA) and compaction of sediments. Various works have investigated the geological and tectonic evolution of the area (Pieri and Groppi, 1981; Ricci Lucchi et al., 1982), the effects of GIA (Lambeck et al., 2011), and the quantification of subsidence in the Po Plain and in the Northern Adriatic coast (Carminati and Di Donato, 1999; Gambolati et al., 1999; Antonioli et al., 2009; Teatini et al., 2011a). These studies have concluded that the natural subsidence in the study region is dominated by the effect of sediment compaction that, along the costal belt, can reach levels comparable to the anthropogenic component of land subsidence. Furthermore, subsidence of glacial isostatic origin has been shown to have presently a minor role along the E-R coastal plain (Stocchi and Spada, 2009).

The anthropogenic subsidence is a consequence of land use and soil exploitation that followed the second World War and is mainly caused by water pumping and gas extraction. These activities have produced a rapid increase of ground subsidence, with rates sometimes exceeding the natural values by one order of magnitude. For example, in the municipality of Ravenna, the observed rate of subsidence rose from 5 (pre-war epoch) to ∼50 mm yr$^{-1}$ (after the war). A detailed discussion of the subsidence connected with the extraction activities can be found in Barends et al. (2005), Teatini et al. (2005) and Arpa Ingegneria Ambientale (2008). At present, following the reduction of most of the water extraction activities, the extraction induced subsidence is mainly attributable to gas fields (Angela-Angelina, see Figure 1), which are still in production. A comprehensive map of the expected regression for the Northern Adriatic coastline at 2100 was published by Gambolati et al. (1998). They also estimated the extent of the floodable area in consequence of the combined effects of subsidence and of a 50 cm increase in sea level, based upon a coarse DTM dataset (see Figure 1.18 in Gambolati et al., 1998).

## 2.2 Sea-level variability

During last millennia, the history of the E-R coast, as well as the Adriatic Sea, has been characterised by a significant sea-level rise and variability. In response to the rapid sea-level rise following the end of the last ice age, in the early Holocene the shoreline experienced a rapid migration from the current latitude of Pescara to approximately the present-day position of the coastline (Correggiari et al., 1996) (see Figure 1). This was followed by a stabilisation of the shoreline close to its present





position, although it has been recognised that, during the post-glacial period, small changes in the climate and in the eustatic component of sea level occurred at scales of decades and centuries (Bruckner, 1890; Friis-Christensen and Lassen, 1991). The migration of the shoreline toward the current position initiated between $4,000$ and $5,000$ years ago, driven by the progradation of the Po river delta and of the beach ridges. During this period, a slow subsidence was acting against the retreat of the sea

water (Correggiari et al., 1996).

At time scales of decades to centuries, changes of the shoreline have been more complex but of smaller amplitude, with shifts up to a few kilometres, mainly driven by the dynamics of rivers (Calabrese et al., 2010) and connected with the ongoing climate change. In this epoch we remark the Medieval Climate Optimum (Veggiani, 1986) with a relative maximum in sea level (IX-X century) followed by the Little Ice Age (XVI-XIX century) with larger sedimentary production and a marine

regression (Marabini et al., 1993; Brázdil et al., 2005). In the Northern Adriatic Sea, the effects of GIA on relative and absolute sea-level change are presently relatively minor, *i.e.* of a few fractions of millimetres per year (Stocchi and Spada, 2009; Galassi and Spada, 2015). Future rates of GIA-induced sea-level variations across the whole Mediterranean Sea (Galassi and Spada, 2014) will not change with respect to current trends since this phenomenon only evolves on time scales of millennia.

### 2.3   Storm surges

The entire northwestern coast of the Adriatic Sea is exposed to a high degree of inundation risk by exceptional sea level caused by storm surges. The E-R coastal plain can be considered a low-energy environment with significant wave height $H_{sig} = 0.4$ m (period $T_{peak}$= 4 s) and semidiurnal and micro tidal regime (spring tidal range = 0.9 m). Beside the meteo-marine component, the variation of sea level is also caused by the astronomic tide, with an average maximum excursion of 0.7-0.8 m (IDROSER S.p.A., 1996; Harley et al., 2012). Storm waves are characterised by a significant height, with $H_s = 3.3$ m for 1 year return

period (Armaroli et al., 2009).

Sea-storm from N and NE are most frequently associated with the Bora wind. South-easterly wind (Scirocco) is a further cause of surge events with wind pushing water up along the coast. The comparatively small strength of Scirocco winds and the sheltering produced by the Conero Headland ($\sim$100 km to the South, see Figure 1) make these events not as intense as those generated by Bora winds (Deserti et al., 2006). A detailed analysis of the storm surge and of its components was carried out by

Masina and Ciavola (2011) using the tide gauge data from Porto Corsini (RA). Their analysis has identified extreme levels of 0.85, 1.05 and 1.28 m for a return period of 2, 10 and 100 years, respectively. The corresponding non-tidal residuals, computed for the same return periods, is 0.61, 0.79 and 1.02 m; such values are in use in Emilia Romagna Region for the computation of total water in the flooding scenarios used for the FD maps (Perini et al., 2012).

In the occurrence of storm surges, the sea level at the coast can vary significantly with respect to the predicted combination

of tide and meteo-marine wave. This is consequence of the local morphology and of the type of the specific waves acting along each portion of the coast. A statistical analysis of the comprehensive dataset of sea storms during the period 1946-2010 (Perini et al., 2011) allowed the definition of threshold values for waves and tides above which significant events, in terms of effects on population, landscape and urban structures, were recorded (Armaroli et al., 2012). These thresholds are in use in the "sea storm alert" procedure adopted by the E-R Regional Civil Defence Agency since 2012.



The stronger impacts are those associated with the coupling between winds from the first quadrant (N and NE) and exceptional tide peaks, known as *"acqua alta"* (Pirazzoli, 1981). The latter, indeed, appears to be crucial for the occurrence of the coastal flooding (Perini et al., 2011). During the observation period 1946-2010, the most critical season has been the late autumn (November and December) while in the last five years these events have been recorded even in February. The most

relevant event was that of November 1966 (Malguzzi et al., 2006; Trincardi et al., 2016), which impacted the whole Northern Italy (De Zolt et al., 2006). This boosted the completion of most of artificial defences currently in place along the E-R coastal plain. With respect to the damages, these events commonly affect the touristic infrastructures along the coast (*e.g.,* the resorts located along the beaches) in consequence of the combination of flooding and of beach erosion. In some cases, the effects are exacerbated by the overflow of the rivers and channels whose discharge is obstructed by the storm surge, causing flooding in

the surrounding urbanised areas and consequently increasing the impact on infrastructure and population.

## 3  Data and Methodologies

The purpose of this work is to analyse the impact of the predicted relative sea-level change at 2100 along the E-R coastal plain. Two distinct aspects will be modelled and discussed. The first is the increase in the extension of land with elevation below sea level and possibly of submerged areas, while the second is the effect of storm surges in terms of floodable areas. In both

cases, the combined effects of sea-level rise and subsidence will be properly taken into account. In this Section, we describe the sea-level projections that have been employed in this work (3.1), some fundamental assumptions in modelling (3.2), and the details of the models adopted (3.3).

### 3.1  Projected sea level

For an assessment of future sea-level rise, we adopt the Representative Concentration Pathways (RCP) scenarios reported in

the IPCC AR5 (Church et al., 2013). By the Coupled Model Inter-comparison Project Phase 5 (CMIP5), the IPCC AR5 has defined RCPs that account for the evolution of the climate variables, and in particular of the $CO_2$ concentration. The RCPs are used for climate modelling to describe four distinct future climate scenarios, characterised by different amounts of greenhouse gas emission. The four RCPs 2.6, 4.5, 6, and 8.5 are named after the possible range of radiative forcing values to year 2100 relative to pre-industrial values, *i.e.* +2.6, +4.5, +6.0, and +8.5 W m$^{-2}$ (see Van Vuuren et al., 2007; Clarke et al., 2007;

Fujino et al., 2006; Riahi et al., 2007, respectively).

The global mean sea-level rise predicted for the four RCPs to 2081-2100 with respect to 1986-2005 is given in Table 2, whereas the projected sea-level rise is shown in Figure 4. Table 2 also shows values of sea-level rise expected across the Adriatic and the Mediterranean Sea, obtained by averaging the AR5 sea-level rise grids downloaded from the Integrated Climate Data Center (ICDC)[7]. We note that values of sea-level rise expected across the Adriatic Sea are systematically lower

than those relative to the Mediterranean Sea. These, in turn, do not exceed the globally averaged values. Therefore, using

T2
F4

---

[7]See goo.gl/QGV5md.



globally averaged values for the IPCC projections like in Antonioli et al. (2017) would overstimate, along the E-R coast, the total amount of sea-level rise.

To focus on the Northern Adriatic Sea, we have extracted from the global AR5 maps the sea-level projections for the grid cell closest to the E-R coast, namely the one centred at latitude $44.5°$ N and longitude $13.5°$ E (see Figure 1). For any RCP, the E-R coast values is found to be slightly lower than those pertaining to the whole Adriatic Sea. We are aware that in semi-enclosed basins like the Mediterranean Sea the ocean model component of sea-level rise could be affected by a limited precision, since the number of models contributing to the "ensemble mean" in these boxes is sub-optimal (Mark Carson, personal communication, 2013). However, we have kept these local projections since the members of the ensemble used to obtain them provide predictions that are broadly consistent with those in the nearby Atlantic Ocean boxes. In the Adriatic Sea cell considered, the assessed GIA component of future relative sea-level rise is $-0.022 \pm 0.005$ m to year 2100 with respect to 1986-2005, where the uncertainty arises from the different predictions of two independently developed GIA models[8].

## 3.2 Assumptions

The phenomena associated with long-term sea-level rise depend on geodynamic, morphodynamic, hydrodynamic and sedimentary factors. In consideration of the challenge of creating a model that accounts for all the aspects of the problem and to limit the computational burden, we have decided to take into account only the changes in the topography due to subsidence, which is the best monitored driving process along the coast. Moreover, several assumptions are necessary to control possible sources of uncertainty, as well as to ease the interpretation of the results.

Specifically, in the process of modelling the effects of sea-level rise to year 2100 (case study CS1) it will be assumed that: *i)* no human intervention, such as for example the reconstruction of the dunes system and the build up of artificial barriers, occurs (no action hypothesis); *ii)* the local rates of subsidence will remain constant over time and equal to those currently observed; *iii)* changes in the coast morphology will be only a function of the rates of subsidence (this implies, in particular, that reshaping of the beaches in consequence of morpho-dynamic processes will also be neglected).

In case study CS2, a meteo-marine scenario will be considered, combining the effects of wave and storm surge. In addition to the three assumptions adopted for CS1, in CS2 it is assumed that *iv)* the meteo-marine conditions at 2100 are unchanged with respect to the present ones. It is worth to remark that diverging future predictions for the rate of storm surges expected at along the E-R coastal plain have been proposed, based on different analyses. In the framework of the MICORE project[9], and by means of time series analysis, Ciavola et al. (2011) have evidenced a tendency to an increase in the rate of storm surges over time, with an unvaried storminess. Besides, by a different approach based on climate models, Lionello et al. (2012) have predicted a decrease in the storminess.

---

[8]See the README.txt at *ftp://ftp.icdc.zmaw.de/ar5_sea_level_rise/*.

[9]See *http://www.micore.eu/*


## 3.3 Modelling

The modelling is based on a GIS cartographic system exploiting the regional database. This provides a quick sketch of the apparent effects of the sea-level rise, highlighting those areas where the impacts could be more critical. To describe the land surface, we have made use of a digital terrain model (DTM), incorporating LIDAR (Light Detection and Ranging) renderings

and a Digital Vertical Movement Model (DVMM), at various spatial resolutions. A comprehensive summary of the datasets used is gathered in Table 3 and described below.                                                                                             T3

### 3.3.1 Case study CS1

In case study CS1, for the determination of the areas with elevation below the mean sea level, the datasets in Table 3 have been merged in a DTM with a $5 \times 5$ m planimetric resolution (DTM2012-RER). This has been preferred to a more detailed and

complex Digital Surface Model (DSM) that incorporates LIDAR renderings including infrastructures, building and vegetation. Specific tests have been performed, which have confirmed that the former provides more realistic results. The DVMM is the result of the interpolation of data acquired by means of PS-InsSAR[10] (Arpa-RER, 2012). This DVMM has been used to project the expected subsidence at 2100 on a $5 \times 5$ m grid. The likely total vertical displacement of the coastal plain to year 2100 is obtained by multiplying the rate by the length of the time interval considered in this study (85 years), and assuming

that subsidence values are constant and equal to those currently observed. The DTM2100 has been obtained adding the total displacement to the DTM2012-RER.

In the framework of CS1, the sea-level variation $S$ to year 2100 is expressed as the combination of three components:

$$S^{CS1}(\omega) = S_{RCP} + S_{GIA} + S_{SUB}(\omega), \tag{1}$$

where $\omega = (\theta, \lambda)$ with $\theta$ and $\lambda$ denoting colatitude and longitude, respectively, $S_{RCP}$ is the contribution according to a specific

RCP, $S_{GIA}$ is the GIA contribution, and $S_{SUB}$ accounts for the effects of subsidence. Note that $S_{GIA}$ and $S_{RCP}$ are assumed to be constant for the study area, while $S_{SUB}$ is spatially variable along the E-R coast (see Figure 3).

Starting from Eq. (1), two different scenarios have been defined. The first, referred to as the WORST case scenario, corresponds to the $S_{RCP}$ value associated with RCP8.5 plus $1\sigma$ (*i.e.*, 0.57 m). The second, referred to as the BEST case scenario, is based on RCP2.6 minus its $1\sigma$ (*i.e.*, 0.23 m, see Table 2). For both scenarios, the contribution of subsidence is

$$S_{SUB}(\omega) = r_{SUB}(\omega)\Delta t, \tag{2}$$

where $r_{SUB}(\omega)$ represents the rate of subsidence at a given location (in meters per year) and $\Delta t$ is the elapsed time span in years. Furthermore, according to the IPCC AR5 assessment (see Section 3.1 above), the projected GIA component is

$$S_{GIA} = -0.022 \pm 0.005 \text{ m.} \tag{3}$$

Thus, Eq. (1) gives

$$S_W^{CS1}(\omega) = 0.55 \text{ m} + S_{SUB}(\omega) \tag{4}$$

___
[10]goo.gl/OQZE1a

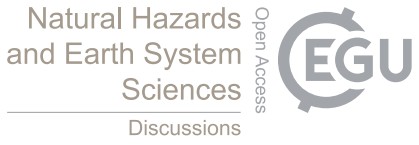

and

$$S_B^{CS1}(\omega) = 0.21 \text{ m} + S_{SUB}(\omega), \tag{5}$$

for scenarios WORST and BEST, respectively.

### 3.3.2 Case study CS2

The objectives of CS2 are *i)* the determination of the potential floodable areas to year 2100 in the occurrence of storm surges
and *ii)* the comparison of these results with the hazard map for the seawater flooding processed in the framework of the
FD (European Parliament and Council, 2007). This topic has a crucial impact in the Emilia Romagna region, in view of the
continuous loss of natural areas and the strong impact that a possible flooding may have on urban areas and on the tourism
infrastructures. This methodology, which follows the approach adopted for CS1, has been based on the modelisation of the
future topographic surface and on the simulation of the flooding in consequence of sea-level rise in scenarios BEST and WORST
described above.

To effectively represent the impact of storm surges to year 2100, the sea-level rise $S(\omega)$ predicted by Eqs. (4) and (5) is
combined with the typical rise $S_{STS}$ associated with characteristic storms in the region, *i.e.,*

$$S_{W,B}^{CS2}(\omega) = S_{W,B}^{CS1}(\omega) + S_{STS}, \tag{6}$$

where $S_{STS}$ is the total sea water level at the shoreline during sea storms, calculated combining the effects of surge, of the
astronomical tide and wave set up. For the $S_{STS}$ component of sea-level change, three scenarios have been adopted in order to
process the hazard maps as requested by the FD. Each of them is associated with a different return period: Frequent (10 years
return period, P3-FD), Less Frequent (100 years, P2-FD), and Rare ($\gg 100$ years, P1-FD, see Table 4) (Perini et al., 2012,   T4
2016; Salerno et al., 2012).

This study is based on the GIS model `in_CoastFlood`[11] developed by *Servizio Geologico, Sismico, e dei Suoli* of Emilia
Romagna Region (SGSS-RER) (Perini et al., 2012), which merges the sea surface in the different weather conditions with the
land topography (DTM lidar). A damping was applied as a function of the distance from the shoreline, with the exception of
those areas which are not directly connected to the sea (Perini et al., 2012; Salerno et al., 2012). Different tests have been
applied to validate the model and to choose the most appropriate scenario. This resulted to be P2, the one with a return period
of 100 years (a standard in the field of country planning) and characterised by more robust reference values. Scenario P3 was
discarded since it affects only a limited portion of the coast, while P1 was discarded because its return period ($\gg 100$ years) is
not clearly defined from a statistical standpoint (Perini et al., 2016).

Once the reference values have been selected (P2-FD), some local tests have been performed to verify if short-term scenarios
are worth to be considered, as they are mainly used in urban planning. The tests have confirmed that short-term scenarios
(*e.g.,* P2-FD in combination with the pessimistic scenario WORST at year 2030) have low significance in terms of change
of the floodable areas when compared to those reported in map for P3 at present, and as a consequence short-term effects

---

[11]See *goo.gl/qxVAWY*

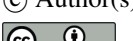



were discarded. Then we used the most detailed DTM available (DTM LIDAR by ENI, 2012) whose resolution ($1 \times 1$ m) exceeds the one used for CS1. The choice of this different dataset was motivated by its higher resolution, necessary to run the `in_CoastFlood` model. The importance of high resolution data and models has been also emphasised by Wolff et al. (2016), who first downscaled the global DIVA approach to the case study of the E-R coastal plain. The model has been used to quantify the change in floodable areas by storm surges at 2012 (P2DTM12, present scenario) and 2100 (P2DTM2100, future scenario).

To optimise the computations and for ease of understanding, the simulation has been performed across five distinct areas, shown in Figure 5. From South to North, the areas range from Cattolica to Rimini (Area 1), from Rimini to Cesenatico (2), from Cesenatico to Fiumi Uniti (3), from Fiumi Uniti to Porto Garibaldi (4), and from Porto Garibaldi to Gorino (5). In Section 4.3 the results are gathered to provide a comprehensive overview at the full E-R regional scale.

## 4 Results

### 4.1 Results for CS1

Aim of this Section is to determine those areas whose elevation at 2100 would move from above to below sea level, by means of realistic estimates of the relative height of the ground. Henceforth we consider the WORST scenario, which implies to a relative sea-level rise of about 55 cm to year 2100.

A first analysis has been performed considering the E-R portion of the Po plain, having an extension of $\sim 9,300$ km$^2$. On this basis, we have computed the variation of land with height above mean sea level by intersecting of DTM2100 with the sea level at 2100, obtained by shifting the present chart datum by $+55$ cm (see Figure 6). As summarised in Table 5, the effect of subsidence would contribute to an emerged land loss of $\sim 101$ km$^2$, whereas considering the combined effect of subsidence and sea-level rise (WORST scenario) the extension of the areas below sea level could reach $\sim 346$ km$^2$. It is apparent that the difference is only restricted to the coastal plain and that large discrepancies exist between the N and the S sectors of the E-R coast. For this reason, we opted for a more detailed analysis by splitting the E-R coastal plain, whose total area is $\sim 3,043$ km$^2$, into three sectors that differ for the morphology and for the response to the scenarios, *i.e.*, the Ferrara, Ravenna, and Rimini sectors. Figure 6 shows, in the WORST scenario, the extent of the areas that will be located below sea level to year 2100 (red). In the same Figure, the areas that today are below sea level are marked in yellow. The largest portion of potentially low land areas to year 2100, is limited to the Ravenna district (right frame in Figure 6). This is in consequence of the high rates of subsidence observed in the area, which is characterised by a low elevation above sea level (see Table 6). It is important to underline that the submerged areas mostly consist of land not directly connected to the sea ("low land areas" or polders).



## 4.2 Results for CS2

The maps obtained for CS2 (see Figures from 7 to 11) show a large increase in the risk of flooding during sea storms along the whole E-R coast, in particular in Area 3, from Ravenna to Cesenatico. For the Areas 1 to 5, we now describe in detail the increase of the floodable areas in case of storm surge with respect to the current scenario at 2012 (P2-DTM2012).

### 4.2.1 Area 1

In Area 1 (Figure 7), the effects of subsidence are found to be very small in comparison with other areas; they result in an increase of the floodable area of $+47\%$ with respect to the one predicted by model P2-DTM12. In the BEST and WORST sea-level scenarios to year 2100, these values increase to $+85\%$ and $+141\%$, respectively. Remarkably, the area mostly affected by the sea-level component is the center of the town of Rimini, but also the mouths of the rivers are impacted. In Area 1, the extension of the floodable area is limited to 50-100 m landward with respect to that predicted by P2-DTM12. F7

### 4.2.2 Area 2

The results for Area 2 are shown in Figure 8. Compared with other areas, Area 2 is characterised by smaller rates of subsidence, smaller depressed regions and higher average elevations above the sea level, which naturally mitigates the combined effect of subsidence and of sea-level rise. The increase of the areas exposed to potential flooding to year 2100 is limited to $+65\%$ with respect to present for the only effect of subsidence, while for the BEST and WORST scenarios we obtain $+102\%$ and $+167\%$, respectively, also including the effects of land subsidence. From the map in Figure 8, we can recognise that from the N border to the Uso river mouth, the extension of the area at risk of flooding is dominated by the effects of subsidence. South of the Uso river mouth, the opposite occurs, with a dominating effect of sea-level rise especially for the WORST scenario. We note that some large portions of urban areas are involved, especially in the municipality of Bellaria-Igea Marina and north of the Rimini harbour. F8

### 4.2.3 Area 3

Area 3 (Figure 9) turns out to be the most critical of the E-R coast in consequence of the high rate of subsidence that characterises its northern portion, between Fiumi Uniti and Cervia. Here we observe the most significant increase ($+248\%$) for the floodable area to year 2100 with respect to 2012 (P2-DTM12), as a consequence of current subsidence rates. When the expected sea-level rise is also taken into consideration, the increase rises up to $+308\%$ and to $+404\%$ for the BEST and the WORST scenarios, respectively. These values evidence that the effect of the sea-level rise is marginal with respect to subsidence, at least for scenario BEST. However, it should be remarked that also in this area the effect of the storm surge would flood the entire coast with a frontal ingression that could extend several hundred meters, exceeding up to 0.5 km the present scenarios, involving large urban areas. F9



### 4.2.4 Area 4

In Area 4 (Figure 10), the scenarios predicted to year 2100 are sharpened by the presence of wide low-elevation regions, with F10
some of them currently below the mean sea level. This is exacerbated by two factors. The first is the total absence of natural
or artificial defences such as dunes and embankments that could counteract the marine action. The second is the high rate of
subsidence, reaching 15 and even 20 mm yr$^{-1}$ at some locations like Porto Corsini, Lido Adriano and Lido di Dante. The
increase of the floodable area, only in consequence of the subsidence, would be of ∼153% relative to P2-DTM12. This rises
to +214% and +294% when the predicted sea-level rise at 2100 is taken into account, according to the BEST and WORST
scenarios, respectively. In this model, the ingression would occur along the whole coast, involving urban areas as well as
natural reserves. The expected inner limit for the areas reached by the flooding would move inland by about 1 km with respect
to P2-DTM12. Furthermore, in case of flooding no natural barriers are present. We should remark, however, that the recent
assessment by Arpa-RER (2012) for the subsidence in this area has revealed a strong decrease to values of 6-7 mm yr$^{-1}$. This
suggests that these proposed scenarios could be revised in the near future, and possibly replaced by more optimistic ones.

### 4.2.5 Area 5

The results for Area 5, shown in the map of Figure 11, indicate an increase of ∼59% of the floodable area in case of storm F11
surge with return period of 100 years (P2 in Table 4), when only the effects from land subsidence are considered (see green
coloured areas). This result is crucial, since currently in the Po Delta area the subsidence is assumed almost completely natural
(Teatini et al., 2011b), with rates that locally reach 11 mm yr$^{-1}$. Accordingly, it appears unlikely to expect a reduction, neither
natural nor induced, in the next 100 years. If we include the effect of the predicted sea-level rise at 2100, the percent increase of
the floodable areas rises to 102% and 214% for the BEST and WORST scenarios, respectively (yellow and red areas). It should
be remarked that a large portion of those areas that will become floodable at 2100 are currently protected by embankments.
The above estimates imply that the current artificial defenses would loose their functionality at 2100. Further considerations
relate with the inward displacement of more than 2 km of the floodable area boundaries relative to the P2-DTM12 predictions.
This would in fact largely involve urbanised areas although the propagation inward of the sea would be localised at weak or
low points along the embankments. As a consequence, a constant and efficient maintenance is required for the embankments
to counteract the effects of subsidence and of future storm surges.

### 4.3 A regional synthesis for CS2

Based on the analysis above, the combined effect of subsidence and sea-level rise in terms of increase of the floodable areas at
2100 can be summarised as follows (see Table 7). *i)* Land subsidence plays a non negligible role, leading to an increase of the T7
floodable areas of ∼95% with respect to that predicted by P2-DTM12 along the entire E-R coastal plain to year 2100. *ii)* The
superimposition of the WORST scenario for sea level would enlarge the extension of the floodable areas in case of P2 event by
more than three times (+236%) with respect to P2-DTM12. *iii)* The most critical portion of the E-R coast in terms of increase
in the floodable area is recognised as the central one, corresponding to Areas 3 and 4; in this respect Area 1 turns out to be





the less critical. *iv)* In the E-R coast portion that extends between Casal Borsetti and Cervia (Areas 3 and 4), the increase is dominated by subsidence which tapers the effect related to sea-level rise. *v)* To the North, areas where subsidence dominates are alternated with those in which sea-level has a major role. Those are mainly localised where defences are in place but gaps are present.

For a correct interpretation of our results, it should be remarked that in CS1 we have only considered the combined effect of subsidence and sea-level rise while in CS2 we have also included the effects of storm surges. In both cases, we are omitting the effects of the coastal morfodynamics and of the solid transport by the sea and by the rivers, as well as the anthropogenic factors that could concur in modifying the landscape by the end of the century. These, in some case could contribute to mitigate the effects of the predicted sea ingression.

## 10     5     Conclusions

In this work we have performed a GIS analysis aimed at identifying those areas of the E-R coastal plain that would suffer, by the end of this century, from an increased vulnerability in response to land subsidence and sea-level rise. These will cause the loss of large portions of emerged land, in particular, beaches, wetlands and dunes. While the first would have a relevant impact on the tourism industry of the region, wetlands and dunes would threaten valuable ecosystems, essential for the preservation of

nature and of life. A second major effect would be the enlargement of the portions of land exposed to water flooding in case of storm surges.

The effects of sea-level rise have been modelled in terms of land loss and increase of floodable areas, assuming storm surges with a return period of 100 years. The distribution of floodable areas has been compared with current scenarios designed by the Emilia-Romagna authority in the framework of the DLGS 49/2010, national emanation of the Flood Directive (EU 2007/60).

The high quality of the data sets employed both at regional and local scale and their enhanced spatial resolution have allowed for a detailed analysis, with a focus on the effect of the single components. The eustatic effect coupled with subsidence has, in the context of the E-R plain, a relevant long-term impact, although in different ways along the coast. The subsidence appears to be the most significant cause in some areas, such a the Ravenna district. The combination of a 55 cm increase in sea level (WORST scenario, based on the IPCC AR5 projections) with a widespread subsidence that is expected to affect with different

rates the entire coastal area, results in an increase of about 346 km$^2$ of the land below sea level. The new areas below sea level could be directly connected to the sea and therefore potentially floodable. This would lead to an increase of the former wetlands reclaimed at the beginning of the last century.

In a step forward, we have combined the above results for the BEST (+23 cm) and WORST (+55 cm) sea-level scenarios with the predicted effects of meteo marine phenomena, *i.e.* storm surges, based on the model "in_CoastFlood" with storm

surges scenarios with return period of 100 years. The results have evidenced that: *i)* subsidence and storm surges would lead to an increase from 29 to 59 km$^2$ of floodable areas at 2100; *ii)* when subsidence and storm surges are combined with sea-level rise, the increase of floodable areas is +133 and +236% for the BEST and WORST scenarios, respectively. In our study, we did not consider possible changes in the current observed rates of subsidence, nor any human intervention to adapt the existing

coastal defence to the evolving shape of the coastline. This work, indeed, discusses the scenario of maximum vulnerability for the case of "no intervention", with the aim of defining the most appropriate actions to be applied in the updating of the FD plan, expected for year 2021. Integrating the above results with the flooding maps produced in the framework of the DGLS 49/2010, it has been possible to compare the current situation with possible future scenarios. The methodology implemented, which

demands relatively limited computational resources, will allow new analyses and further maps to plan new possible mitigation actions as well. This would leave however the Emilia-Romagna coastal plain exposed to a certain level of risk, in consequence of its natural shape and geology. In this respect, this work can be considered the starting point for decision makers as well as for the scientific community to define the areas exposed to a major risk, the possible actions to mitigate the predicted risks, and the strategies to handle the forthcoming strong surges and flooding.

**6   Code availability**

The codes may be available on request from LP.

**7   Data availability**

The data may be available on request from LP.

*Author contributions.* The work presented here was carried out in collaboration between all authors. LP, LC and PL defined the research

theme, designed most of the methods and experiments, and wrote the paper. MO, GG and GS have designed some of the methods and experiments, have contributed to the presentation and interpretation of the results, and to the drafting of the manuscript. All authors have contributed to, seen and approved the manuscript.

*Competing interests.* The authors declare no competing interests

*Disclaimer.* To be written

*Acknowledgements.* Sea level projections have been extracted from the Integrated Climate Data Center (ICDC, *http://www.icdc.zmaw.de/*) of the Hamburg University on October 2015. GG and GS are partly supported by research grants of the Department of Pure and Applied Sciences (DiSPeA) of the Urbino University "Carlo Bo" (CUPs H32I160000000005 and H32I15000160001). Some of the figures have been drawn using the Generic Mapping Tools (GMT) (Wessel et al., 2013).



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

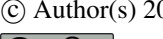


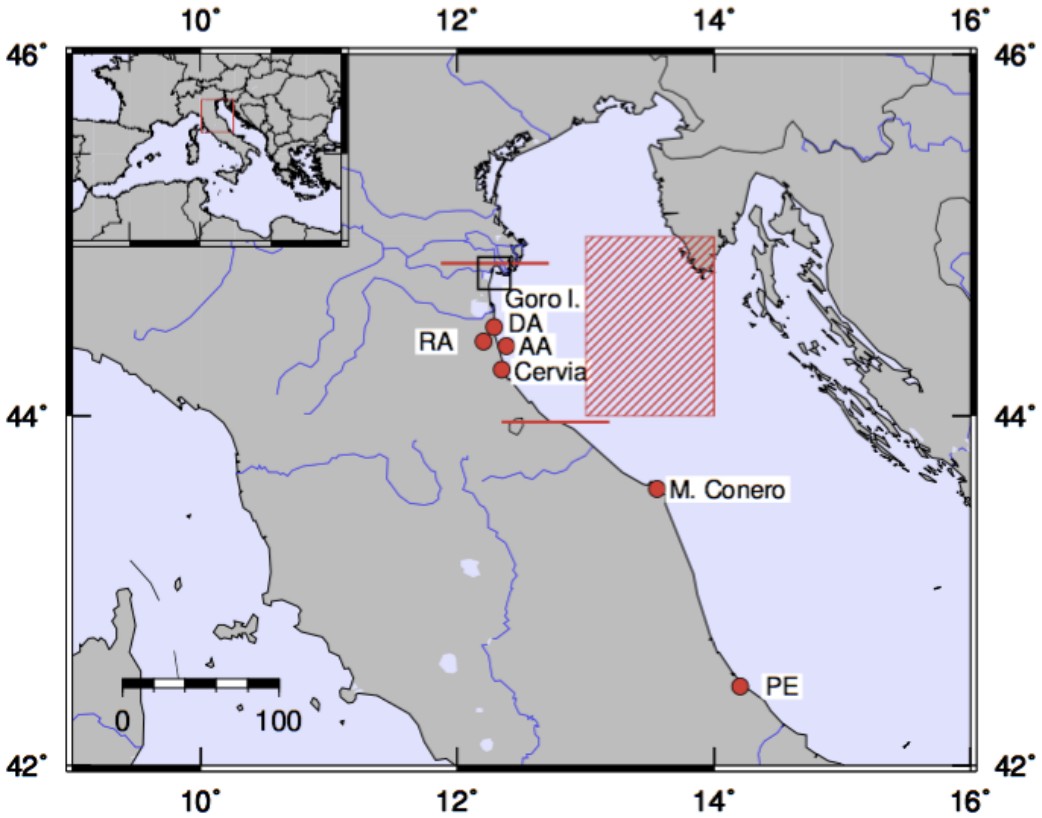

**Figure 1.** Map of the Northern Adriatic Sea and of the surroundings. Horizontal ticks on the East coast of Italy mark the northern and southern limits of the E-R shoreline. The hatched rectangle shows the cell of the IPCC AR5 discretisation of the global oceans used to represent the projected sea-level change in front of the E-R coast (RA: Ravenna, AA: Angela-Angelina gas field, PE: Pescara). The town of Cervia divides E-R coast into two sectors, referenced to as North (N) and South (S) in the body of the paper.





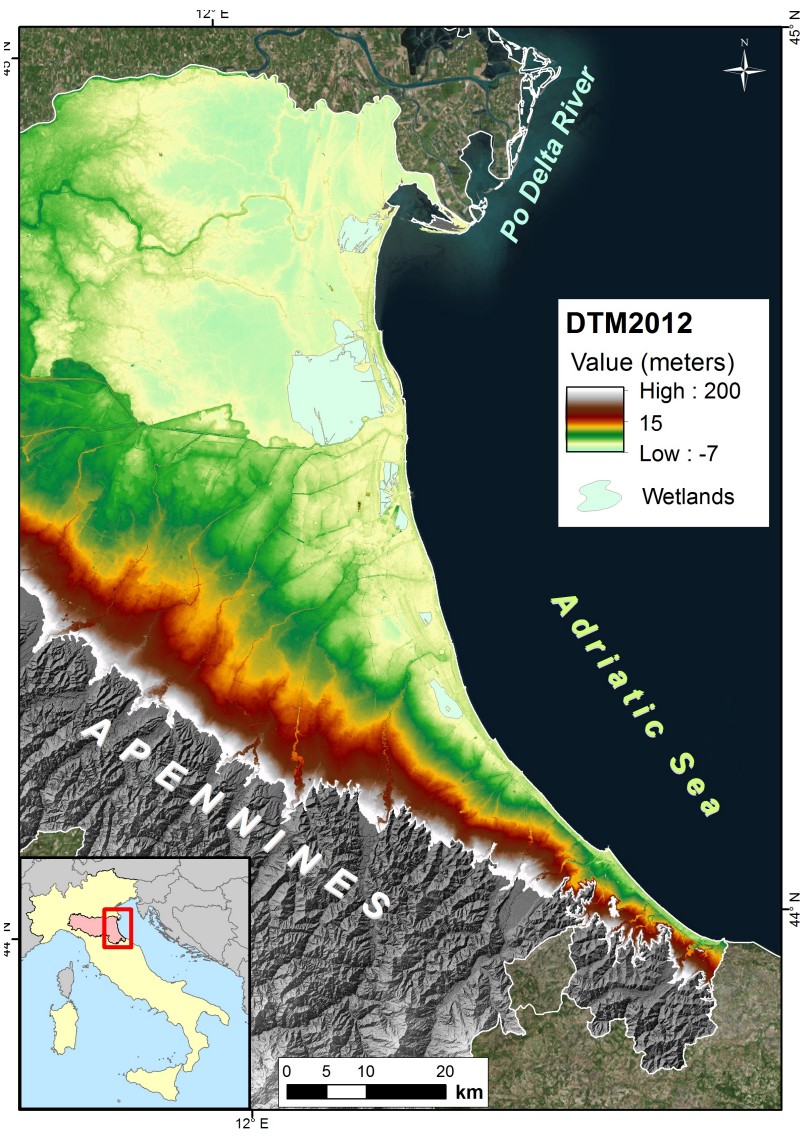

**Figure 2.** The altimetry and morphology of the E-R coastal plain resulting from a high resolution Digital Terrain Model (DTM). The remarkable difference between the N and the S sector in term of the extension of low lands is apparent.




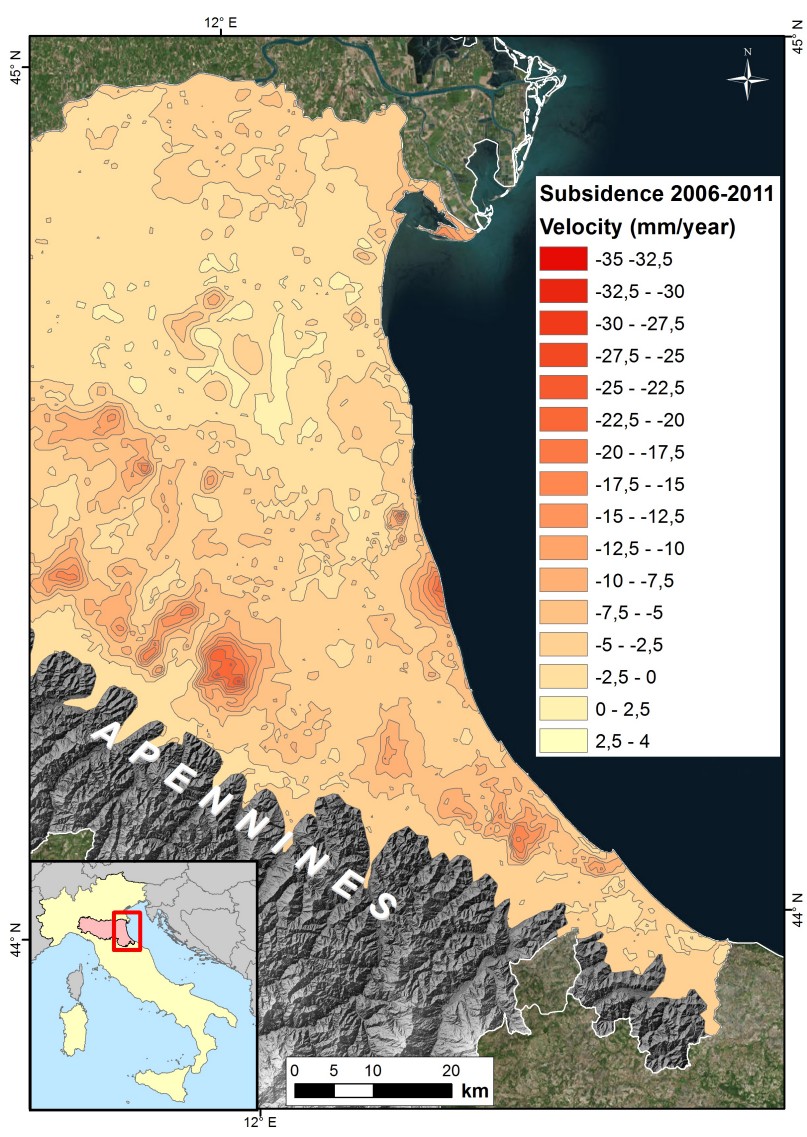

**Figure 3.** Subsidence rate map of the E-R coastal plain according to the InSAR monitoring for the time period 2006-2011 (Arpa-RER, 2012).





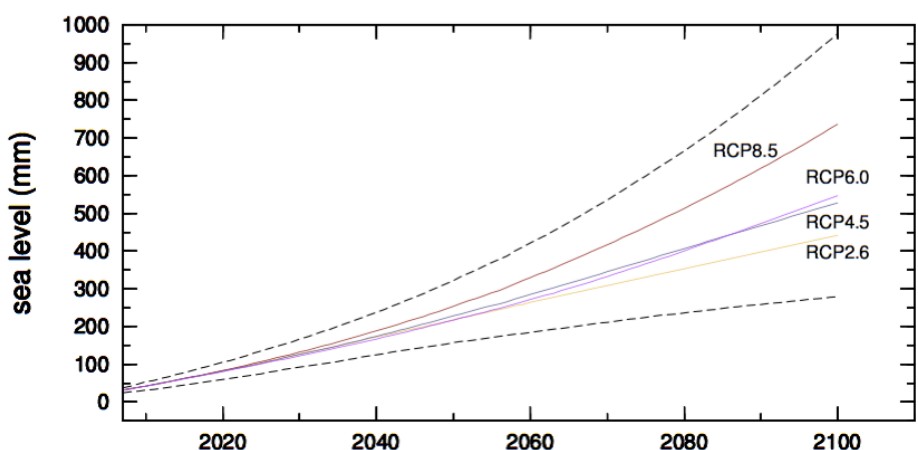

**Figure 4.** Sea-level pathways according to IPCC AR5. Dashed lines show the lower and the upper limits of the projections, corresponding to the upper limit of RCP 8.5 and to the lower limit of the RCP 2.6. Projections are referred to the average sea-level values in the period 1986-2005.




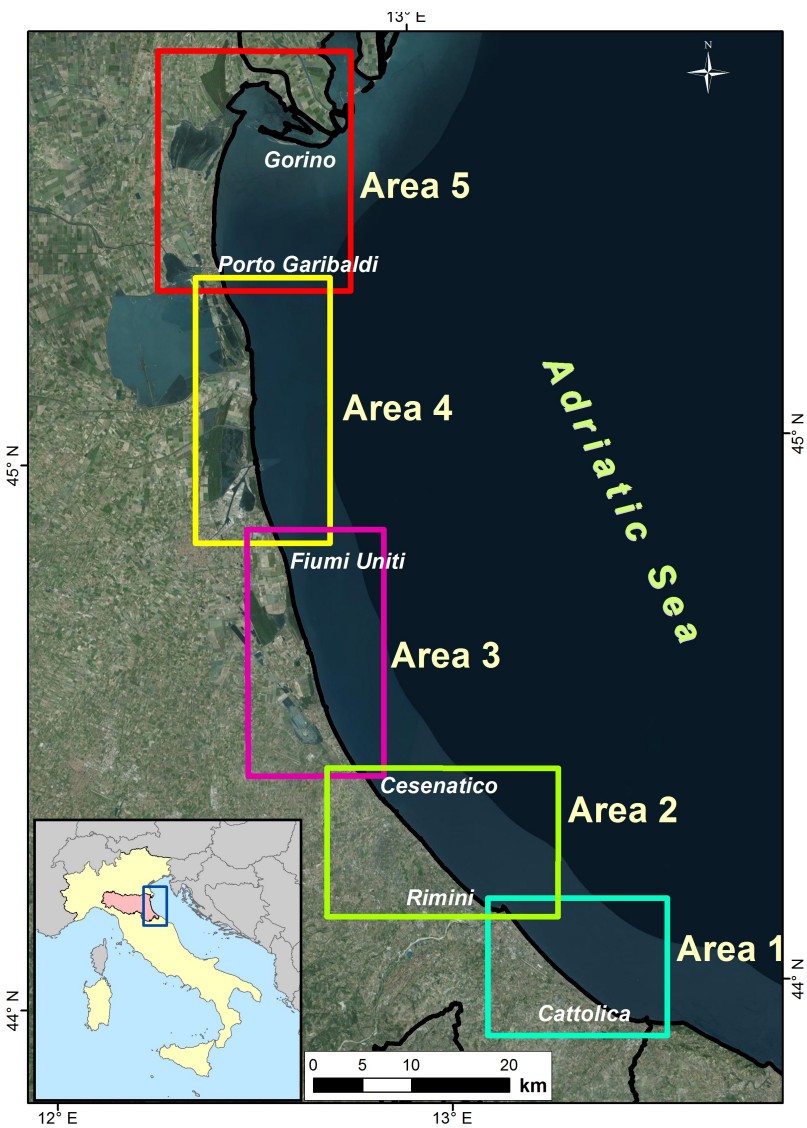

**Figure 5.** Location and boundaries of the five areas (Area 1 to Area 5) on which the E-R coastal plane has been divided.





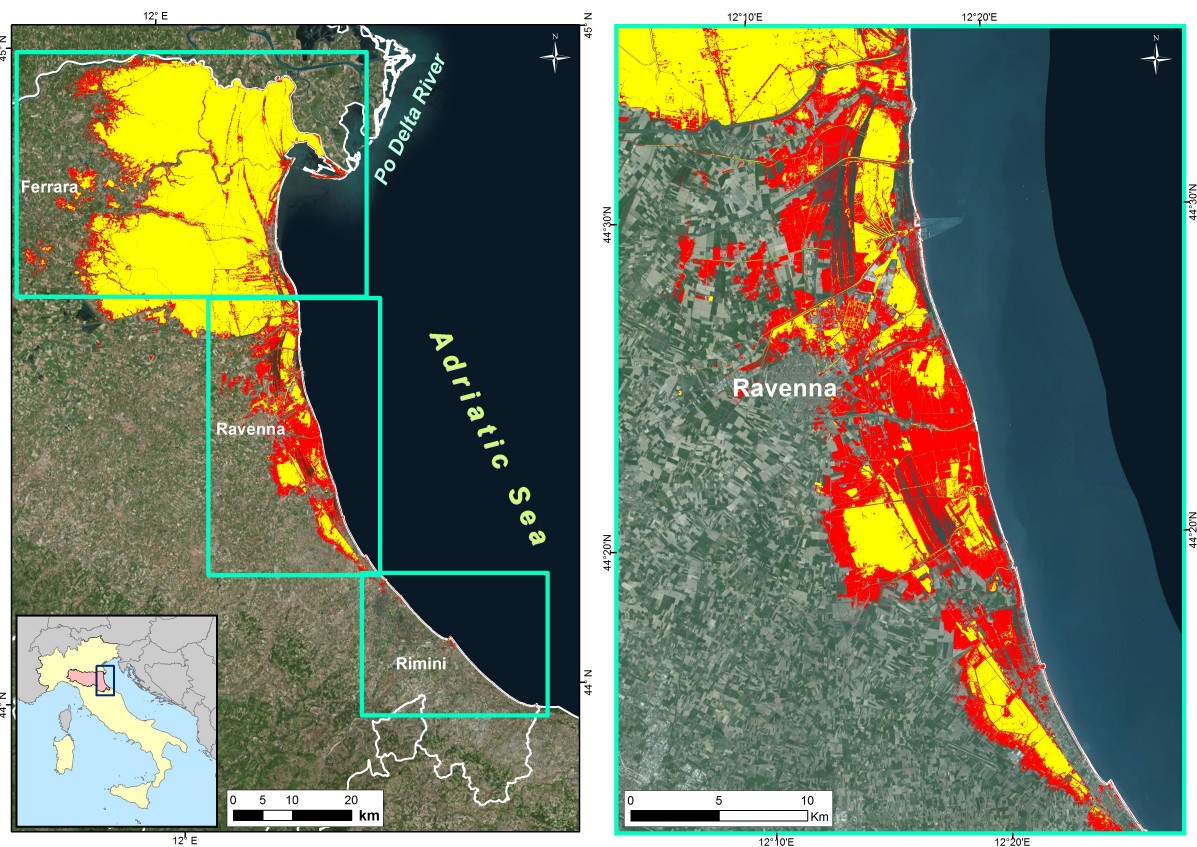

**Figure 6.** Results for the simulations in CS1, at the E-R coastal plain scale (left) and for the area of Ravenna (right). Lands that are currently located below sea level are evidenced in yellow. New areas predicted to be located below sea level at 2100 are marked in red.




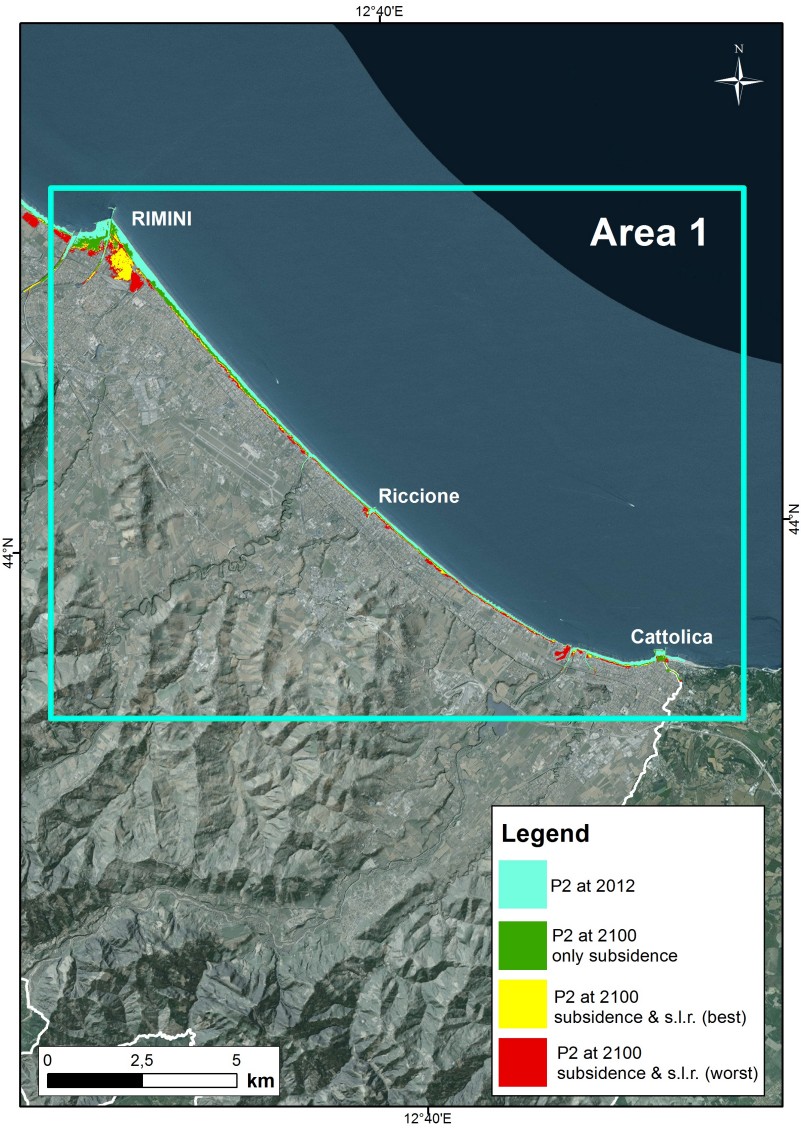

**Figure 7.** Floodable regions across Area 1 in case of storm surge for case P2 at 2100. The assumptions in modelling are color-coded in the inset.





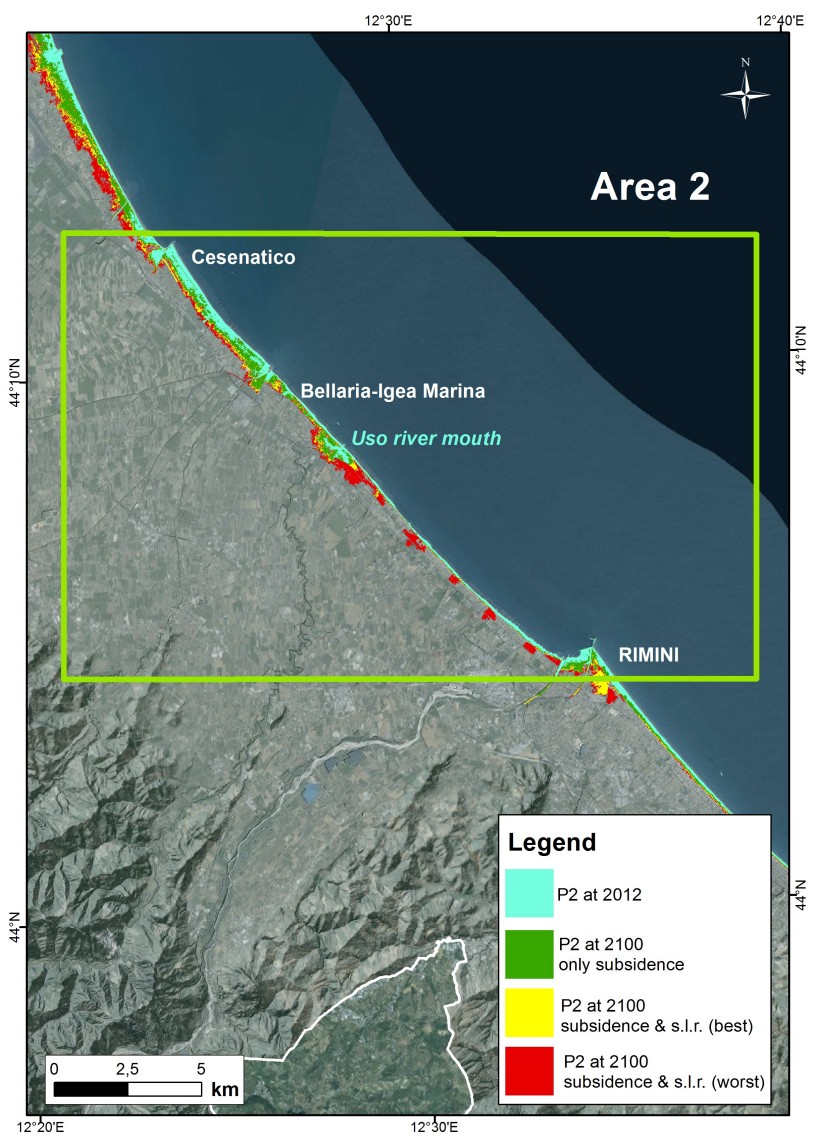

**Figure 8.** The same as in Figure 7, but for Area 2.




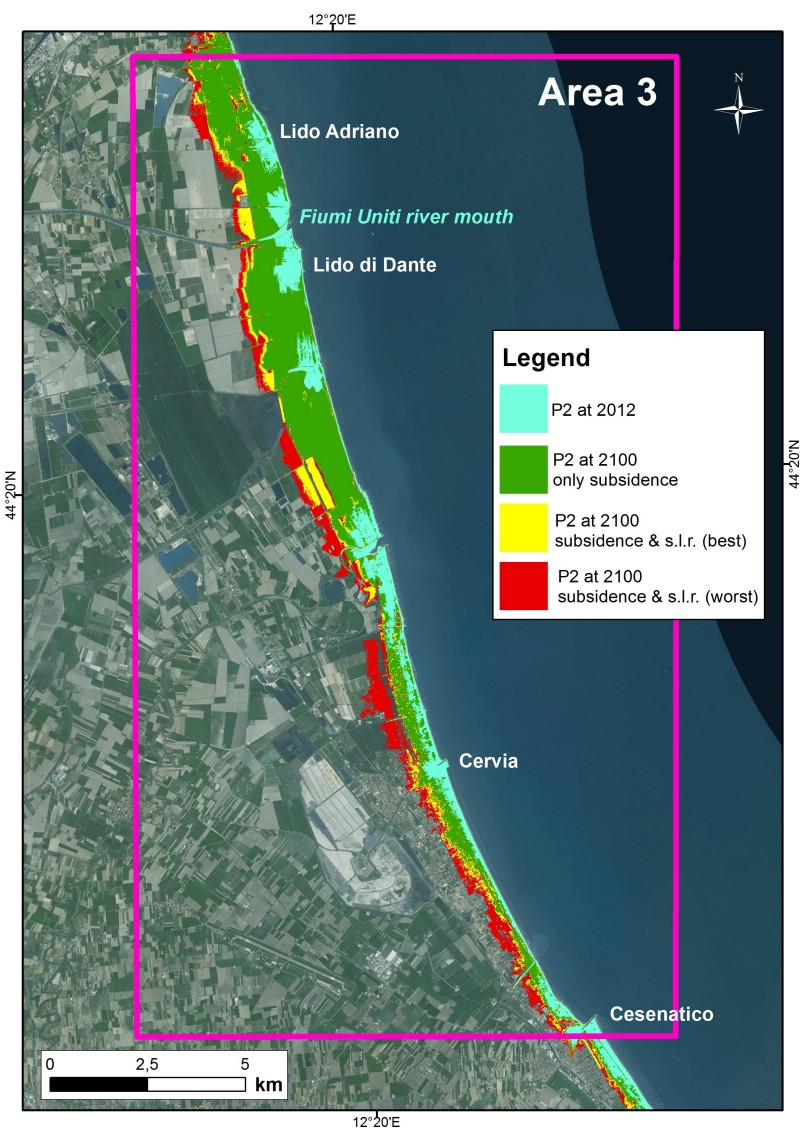

**Figure 9.** The same as in Figure 7, but for Area 3.





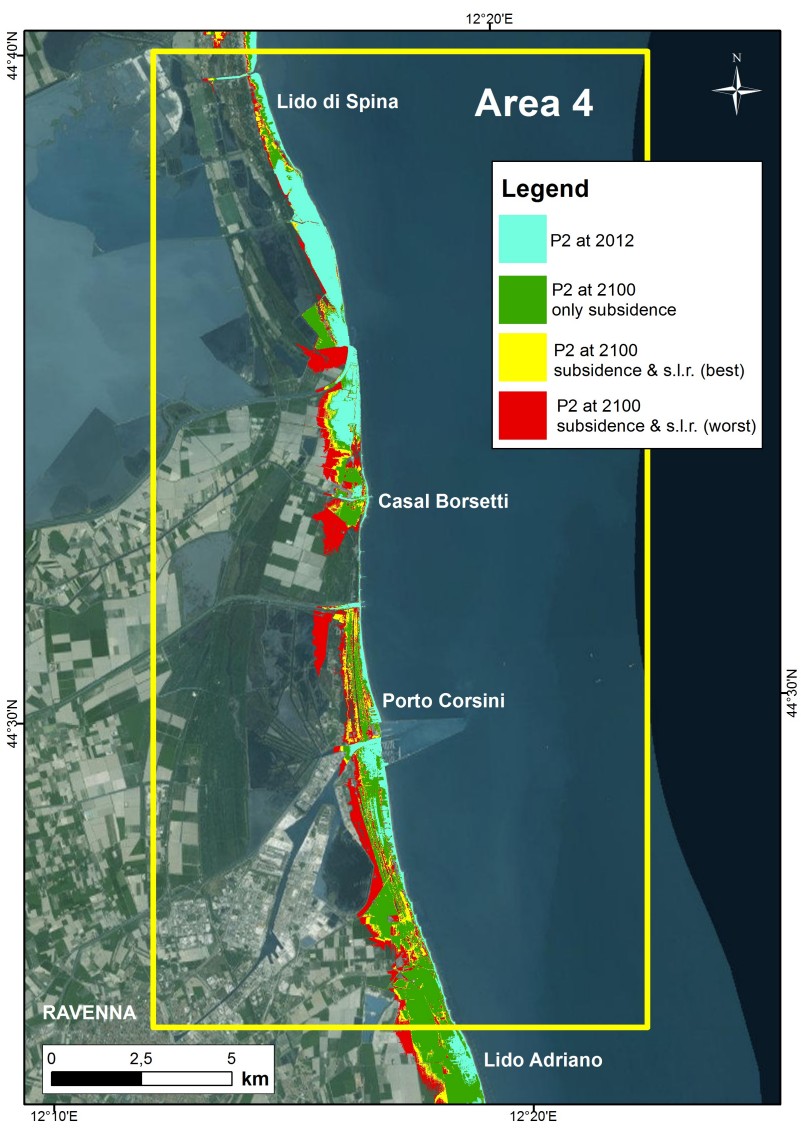

**Figure 10.** The same as in Figure 7, but for Area 4.





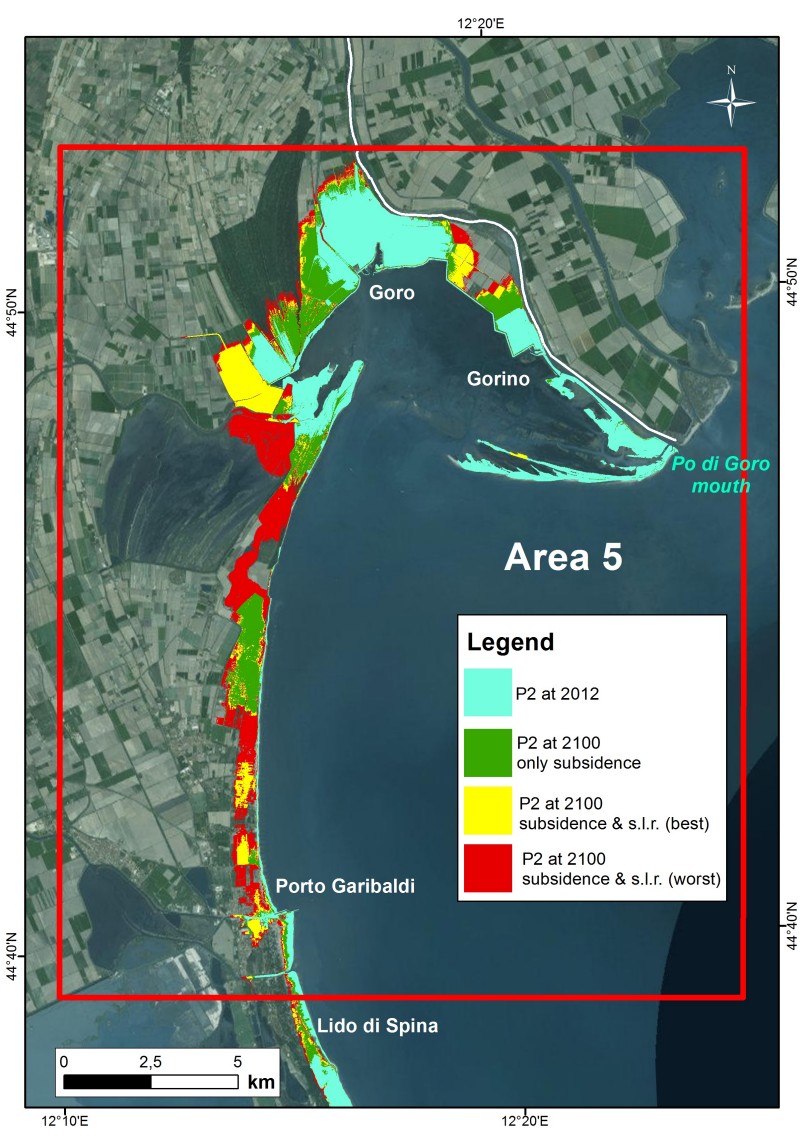

**Figure 11.** The same as in Figure 7, but for Area 5.



**Table 1.** Summary of the physical characteristics of the E-R coast. The N and S sectors are defined in Figure 1.

|  | North sector (N) | South sector (S) |
|---|---|---|
| Coastal plain width ($D$) | $5 < D < 40$ km (from South to North) | $0.8 < D < 5$ km (from South to North) |
| Dominant elevation above sea level ($H$) | $-1 < H < 1$ m | $0 < H < 3$ m |
| Dry beach width ($D_s$) | $0 < D_s < 300$ m, mean value $\overline{D}_s = 60$ m | $10 < D_s < 180$ m, mean value $\overline{D}_s = 80$ m |
| Geomorphological features | Beach ridges, dunes, lagoons, wetlands, reclaimed areas, sand spits | Beach ridges, dunes (only towards the north), saltpans |
| Shape of the shoreline | Wavy, convexities at river estuaries, more complex at river spits | Straight with asymmetric wedges at Cesenatico and Rimini harbors |
| Backshore land use | Urbanized areas, agriculture, vegetation and wetlands | Wide urbanized areas, agriculture and vegetation |
| Medium/long term tendency | Variable, tending to rectify | Stable, with local changes |





**Table 2.** Sea level predicted during the time interval 2081-2100 with respect to 1986-2005, according to the four IPCC AR5 RCPs. The Adriatic and Mediterranean values are averaged values across these seas. These projections do not include the GIA component of sea-level change.

| RCP | E-R coast | Adriatic | Mediterranean | Global |
|---|---|---|---|---|
| | m | m | m | m |
| 2.6 | 0.30 ± 0.07 | 0.31 ± 0.01 | 0.36 ± 0.02 | 0.38 ± 0.15 |
| 4.5 | 0.34 ± 0.09 | 0.37 ± 0.01 | 0.42 ± 0.03 | 0.45 ± 0.16 |
| 6.0 | 0.33 ± 0.08 | 0.36 ± 0.02 | 0.42 ± 0.03 | 0.47 ± 0.16 |
| 8.5 | 0.45 ± 0.12 | 0.48 ± 0.02 | 0.57 ± 0.03 | 0.60 ± 0.19 |

**Table 3.** Description of the four input models employed for the definition of case studies CS1 and of CS2.

| Input dataset name | Interpolated points | Altimetric tolerance | Output dataset name | Grid size |
|---|---|---|---|---|
| LIDAR 2008 PNT | 2 p/m$^2$ | ±15 cm | DTM 2008 PNT | 2 × 2 m |
| LIDAR 2010 RER | 4 p/m$^2$ | ±10 cm | DTM 2010 RER | 1 × 1 m |
| LIDAR 2012 ENI | 4 p/m$^2$ | ±10 cm | DTM 2012 ENI | 1 × 1 m |
| PS-InSAR 2006-2011 PNT | 1 p/26 m$^2$ | ± 2 mm | DVMM | 5 × 5 m |





**Table 4.** Expected sea-level rise in the occurrence of different type of characteristic storm surges and their expected return period (Perini et al., 2012, 2016; Salerno et al., 2012).

| Type of surge | Return period | Sea surface elevation |
|---|---|---|
| | years | m |
| Frequent (P3) | 10 | 1.49 |
| Less Frequent (P2) | 100 | 1.81 |
| Rare (P1) | > 100 | 2.50 |

**Table 5.** Areas expected to be above and below mean sea level (m.s.l.) to year 2100, compared to year 2012. Upperscript [a] denotes only subsidence; [b] includes subsidence and sea-level rise.

| Year | Above m.s.l. | Below m.s.l. | Lost area above m.s.l. |
|---|---|---|---|
| | km$^2$ | km$^2$ | km$^2$ |
| 2012 | 8083.8 | 1216.4 | |
| 2100 [a] | 7982.5 | 1318.3 | 101 |
| 2100 [b] | 7737.6 | 1563.2 | 346 |




**Table 6.** Detail of the future areas below sea level, compared to year 2012.

| Area | at 2012 km$^2$ | at 2100 km$^2$ | Difference km$^2$ |
|---|---|---|---|
| Ferrara | 1143.32 | 1331.41 | 188.09 |
| Ravenna | 72.00 | 224.28 | 152.28 |
| Rimini | 0.72 | 6.34 | 5.61 |

**Table 7.** Regional floodable areas in case of storm surges for case P2 described in Table 4. The increase in amount of flooded area is relative to P2-DTM12. Upperscript [a] denotes only subsidence, [b] includes subsidence and sea-level rise (BEST scenario), [c] includes subsidence and sea-level rise (WORST scenario).

| Case | Flooded area km$^2$ | Increase % |
|---|---|---|
| P2-DTM12 | 29 | |
| 2100 [a] | 59 | 95 |
| 2100 [b] | 72 | 133 |
| 2100 [c] | 105 | 236 |