# Peer review of "Sea-level rise along the Emilia-Romagna coast (Northern Italy) at 2100: scenarios and impacts"

_Natural Hazards and Earth System Sciences, 2017_

## Referee Comment (RC1) · Anonymous Referee #1 · 23 Mar 2017

It is an interesting paper that illustrates the consequences of mean sea level rise and storminess on the coastal areas of Emilia-Romagna in terms of land loss. The paper is logically structured and the methodology seems adequate (but I cannot make comments on the flood model because I don't know it). Unfortunately, the paper often contains confusing terminology that makes it hard to understand. This problem should be solved before the paper can be published.

Main problems.

a) The authors use 'sea level' both for a long term (multi-decadal) mean and a short-term (e.g. hourly) value, as in the case of storm surges. For instance at page 3, line 19, the meaning is 'mean sea level', like at page 9, line 8 (case CS1), where the full 'mean sea level' is used. By contrast, at page 6, line 18, the meaning is 'sea level height'

relative to a known reference. As another example, at page 10, line 5, the authors only consider storm surges, while case CS2 also includes the wave effect. Definitions should be clear and used consistently. Note that Sect. 2.2 should involve the 'mean sea level'.

b) The use of 'wave' is also often unclear. For instance, 'storm waves' at page 6, line 19 are clearly wind waves, while 'meteo-marine wave' and 'specific waves' (same page, line 30) are undefined. Please specify when wind waves or any other wave type are involved.

c) The authors deal with both storm surges and wind waves (case CS2). Sect. 2.3 is devoted to storm surges, but something about wind waves is also included. This is confusing because the role of wind waves is first mentioned explicitly only at page 10, line 16. The authors might consider to deal with storminess in general, that is both storm surges and wind waves (see page 8, line 23), and introduce the subject accordingly.

Specific comments.

Page 1, line 1: Are there any reasons to neglect natural land subsidence?

Page 1, line 13: Please clarify that '(in_CoastFlood)' is the name of the model. The missing reference, the different typos and the brackets are confusing.

Page 4, Lines 15-16: 'some cases . . . sometimes' is redundant.

Pages 5-6, Sect. 2.2: The discussion of mean sea-level changes over thousands of years is not crucial for present and future variations. By contrast, there is no discussion on the mean sea level variations during the last 100 years or so, when also anthropogenic subsidence occurred.

Page 6, lines 17, 30: The exact meaning of 'meteo-marine' is unclear in this context. I guess that the authors mean 'the sea level changes component related to the atmospheric forcing', which includes both wind and atmospheric pressure (not mentioned). This component is the 'residual sea level' also known as the 'meteorological tide'. Moreover, is this the 'non-tidal residual' at line 26?

Page 6, line 27: Please quote the reference to which the reported heights are measured.

Page 6, lines 29-31: Unclear. I understand (but I am not sure) that the observed sea level can differ from the forecast represented by the astronomical tide plus the residual sea level ('meteo-marine wave' is bad terminology). The difference does not occur because of local morphology and specific waves, but because the model used for the predictions is not good enough. For instance, it may not include the correct basin bathymetry and coastal morphology, or the atmospheric forcing is too coarse. Anyway, the sentences can be dropped.

Page 6, line 32: Both waves and tides are mentioned. It is not clear what 'tides' mean here.

Page 7, line 6: Do the authors mean Adriatic instead of Italy?

Page 8, line 5: Unclear sentence. What are 'the E-R coast values'?

Page 9, line 14: The IPCC mean sea level rise projections are made for 2081-2100 (central year 2090.5) relative to 1986-2005 (central year 1995.5) (page 7, line 26), that is a 95-year time period, but the authors use a 85-year period. Is that a mistype?

Page 12, line 9: Is 'sea-level component' quoted in comparison to subsidence? Please clarify.

Page 14, line 29: Does the model include wind waves set up? These are not mentioned here, while they seem to have been at Page 10, line 16, when they are distinct from the surge.

Page 15, line 33: The authors should not only say that subsidence rates are assumed unchanged in the 21st century, but also the storminess characteristic.

Page 21, Fig. 2: Please say if the zero height in the map corresponds to the 1986-2005 mean sea level (the IPCC start time), to the zero of the Italian geodetic network or to another thing. I also suggest to use a colour palette that highlights the altitude differences in the low-lying areas. Probably, a 0 m contour could also be useful.

Pages 26-30, Fig. 7-11: The coloured areas are often small compared to the whole figure and most of them are barely visible. Can the authors improve their visibility?

Page 33, Table 4: In the text (page 10) rare events have a return period »100 years, not >100.

---

## Referee Comment (RC2) · G. Le Cozannet (Referee) · 12 May 2017

The article by Perini et al. provides an estimation of sea-level rise impacts (in terms of erosion and flooding in the Emilia-Romagna region). Interestingly, the article considers regional subsidence patterns, which have a high spatial variability as shown by previous observations based on SAR interferometry. The article also illustrates how the application of European directives can stimulate studies and discussions regarding the future impacts of climate change. Overall, I think that the article provides an interesting perspective, and that it is relevant to NHESS.

What is missing, to my opinion in the article, is a real discussion of the significance of the results and their implications. I can suggest a few recommendations in this respect:
- It could be interesting for the reader to know how such work (which is apparently

strongly connected to regulatory processes such as the European flooding directives, e.g. page 3 line 22) will (or is expected to be) integrated in regional to local adaptation. I have no specific suggestion here, but I just remind that the AR6 IPCC reports to come will require information on the implementation of adaptation (including its successes and limitations). I think that the authors can make a useful contribution here. - The authors clearly list their assumption all along their study (e.g. section 3.2), but the reader would like to see a discussion on the impacts of these assumptions in the final results. I suggest that uncertainties in the results could be given more attention in a discussion section (see below further suggestions). - Finally, if possible, it would be interesting to see to which extent this study agrees or disagrees with previous impacts assessments performed in the same region (e.g. Wolf et al., 2016) and why.

I provide below detailed comments, which are hopefully useful if the authors decide to discuss uncertainties: - Subsidence: I wonder to which extent it is realistic to assume that subsidence is linear in time. In practice, the authors show that it has not been the case in the past (with acceleration in subsidence rates with increased fluid extraction in section 2.1), and this seems to me relatively common in cases of subsidence caused by groundwater extractions (Le Mouelic et al., 2005; Wang et al., 2012; Raucoules et al., 2013). I wonder if the authors would agree that in their table 5, they provide the maximum benefits of an adaptation strategy consisting in mitigating subsidence through reduced fluid extractions. A small point Page 5 line 12: "compaction of sediments" is unclear to me. I assume the authors refer here to natural (and later, anthropogenic) variability of water content in various geological layers, resulting in a reduction of their volume. - Extreme water levels: The authors use value of water heights during storms (subsection 2.3). However, it is unclear which processes have been incorporated. Of course, the references to the project Micore and other studies suggest that tides, atmospheric surge, wave setup (Stockdon et al. 2006) have been taken into account, but I suggest naming these processes explicitly. Note that the wave setup can account for an additional contribution of several 10cm, which is not negligible considering the magnitude of sea-level changes to come. If no information is available on this process, this

source of uncertainties can be assumed dominant for the decades to come. - Mean sea-level projections: Sea-level projections used in this article rely on global models, which have not the ability to represent processes taking place at the Gilbratar straight (subsection 2.4). This can result in deviation of some 10 cm from sea-level projections in the Atlantic, west of the Gibraltar straight. Also, is the area affected by 3D circulation modifying water levels by +/-10cm as it is the case in the gulf of Lion? I suggest to discuss these processes in a discussion on uncertainties. They are discussed for example in Adloff et al. (2015, 2016, both in Climate Dynamics) and also in our article Le Cozannet et al. (2015 in Environmental Modeling and Software). Furthermore, the wording "Worst" or "best" cases scenarios (page 9 line 22 and several times after) is not appropriate for ranges of uncertainties representing likely confidence intervals (see Church et al., 2013a, 2013b) and can be misleading for coastal managers in charge of adaptation (Hinkel et al 2015). This should be rephrased. - Impacts : The authors have presented their results in two ways : "land losses" due to sea-level rise and subsidence (e.g., conclusion) and "areas lying below mean sea-level" (e.g. in table 5). I am personally in favor of the second formulation, as it makes no assumption on the adaptive responses to come (e.g., beach and dunes nourishment. . .). In both cases, the results assume no morphological changes, which, again, would deserve a discussion. There is a huge bibliography in this area. Finally, can the authors explain why storm surge impacts have not been assessed in both CS1 and CS2 hazard assessments (page 14 line 5)? Finally, I wonder if figure 3 and 5 could be merged. I hope these comments are useful.

References Wolf et al. 2016; Church et al., 2013a: see author's reference list Church et al. 2013b: Church J A, Clark P U, Cazenave A, Gregory J M, Jevrejeva S, Levermann A and Payne A J 2013 Sea-level rise by 2100 Science 342 1445 Wang et al., 2012: Wang, J., W. Gao, S. Y. Xu, and L. Z. Yu (2012b), Evaluation of the combined risk of sea level rise, land subsidence, and storm surges on the coastal areas of Shanghai, China, Clim. Change, 115(3-4), 537–558, doi:10.1007/s10584-012-0468-7. Raucoules et al., 2013: Raucoules, D., G. Le Cozannet, M. Gravelle, M. de Michele, G. Wöppelmann,

M. Gravelle, A. Daag, and M. Marcos (2013), Strong non-linear urban ground motion in Manila (Philippines) from 1993 to 2010 observed by InSAR, Remote Sens. Environ., 139, 386–397, doi:10.1016/j.rse.2013.08.021. Adloff F, Somot S, Sevault F, Jordà G, Aznar R, Déqué M, Herrmann M, Marcos M, Dubois C, Padorno E, Alvarez-Fanjul E, Gomis D; Mediterranean Sea response to climate change in an ensemble of twenty first century scenarios. Climate Dynamics, 2015, Volume 45, Issue 9, pp 2775-280258. Adloff F., Jordà G., Somot S., Sevault F., Arsouze T., Meyssignac B., Li L., Planton S. Improving sea level simulation in Mediterranean Regional climate models. Climate Dynamics. (2016) Le Cozannet G, Rohmer J, Cazenave A, Idier D, van De Wal R, De Winter R and Oliveros C 2015 Evaluating uncertainties of future marine flooding occurrence as sea-level rises Environ. Modell. Softw. 73 44–56 Stockdon, H. F., Holman, R. A., Howd, P. A., & Sallenger, A. H. (2006). Empirical parameterization of setup, swash, and runup. Coastal engineering, 53(7), 573-588. Le Mouelic, S., Raucoules, D., Carnec, C., & King, C. (2005). A least-squares adjustment of multi-temporal InSAR data: Application to the ground deformation of Paris. Photogrammetric Engineering and Remote Sensing, 71, 197–204. Hinkel J, Jaeger C, Nicholls R J, Lowe J, Renn O and Peijun S 2015 Sea-level rise scenarios and coastal risk management Nat. Clim. Change 5 188–90

---

## Author Comment (AC1) · 14 Jun 2017

Urbino 15 giugno 2017

Dear Editor, please find below the author's response to the referee comments on paper "Sea-level rise along the Emilia-Romagna coast (Northern Italy) at 2100: scenarios and impacts" by Luisa Perini et al. (ms No.: nhess-2017-82). For your convenience, below the Reviewers's queries have been numbered as Qx.y, where x is the Reviewer (x=1, 2) and y is the point raised. Our response is labeled by Ax.y. We are looking forward to receive your decision about the further handling of the manuscript.

Sincerely yours

Giorgio Spada (on behalf of all co-authors)

[Figure]

Review 1 (Anonymous)

Q1.0 It is an interesting paper that illustrates the consequences of mean sea level rise and storminess on the coastal areas of Emilia-Romagna in terms of land loss. The paper is logically structured and the methodology seems adequate (but I cannot make comments on the flood model because I don't know it). Unfortunately, the paper often contains confusing terminology that makes it hard to understand. This problem should be solved before the paper can be published. A1.0 We thank the Reviewer for his positive response. We acknowledge that the terminology is confusing in some parts of the manuscript, and we shall make efforts to improve it.

Main problems. Q1.1 a) The authors use 'sea level' both for a long term (multi-decadal) mean and a short term (e.g. hourly) value, as in the case of storm surges. For instance at page 3, line 19, the meaning is 'mean sea level', like at page 9, line 8 (case CS1), where the full 'mean sea level' is used. By contrast, at page 6, line 18, the meaning is 'sea level height' relative to a known reference. As another example, at page 10, line 5, the authors only consider storm surges, while case CS2 also includes the wave effect. Definitions should be clear and used consistently. Note that Sect. 2.2 should involve the 'mean sea level'. A1.1 We recognise that the terminology regarding "sea level" may be inconsistent in some cases, and needs improvements.

Q1.2 b) The use of 'wave' is also often unclear. For instance, 'storm waves' at page 6, line 19 are clearly wind waves, while 'meteo-marine wave' and 'specific waves' (same page, line 30) are undefined. Please specify when wind waves or any other wave type are involved. A1.2 We realise that the use of words "wave" and "waves" needs to be checked throughout the manuscript to avoid ambiguous statements.

Q1.3 c) The authors deal with both storm surges and wind waves (case CS2). Sect. 2.3 is devoted to storm surges, but something about wind waves is also included. This is confusing because the role of wind waves is first mentioned explicitly only at page 10, line 16. The authors might consider to deal with storminess in general, that is

both storm surges and wind waves (see page 8, line 23), and introduce the subject accordingly. A1.3 We propose to entitle Section 2.3 "Sea storms", to include both storm surge and waves. Of course, the terminology is to be changed accordingly throughout the paper.

Q1.4 Page 1, line 1: Are there any reasons to neglect natural land subsidence? A1.4 There is no reason to neglect natural land subsidence, effectively.

Q1.5 Page 1, line 13: Please clarify that '(in_CoastFlood)' is the name of the model. The missing reference, the different typos and the brackets are confusing. A1.5 This can be fixed in the revised manuscript, in order to clarify that '(in_CoastFlood)' is the name of the model.

Q1.6 Page 4, Lines 15-16: 'some cases . . . sometimes' is redundant. A1.6 We agree.

Q1.7 Pages 5-6, Sect. 2.2: The discussion of mean sea-level changes over thousands of years is not crucial for present and future variations. By contrast, there is no discussion on the mean sea level variations during the last 100 years or so, when also anthropogenic subsidence occurred. A1.7 Effectively, a discussion on the mean sea level variations during the last century is missing. We propose to add a paragraph on this subject, due to its relevance with the topics dealt with in the paper.

Q1.8 Page 6, lines 17, 30: The exact meaning of 'meteo-marine' is unclear in this context. I guess that the authors mean 'the sea level changes component related to the atmospheric forcing', which includes both wind and atmospheric pressure (not mentioned). This component is the 'residual sea level' also known as the 'meteorological tide'. Moreover, is this the 'non-tidal residual' at line 26? A1.8 We agree on the need of better specifying the exact meaning of "meteo-marine".

Q1.9 Page 6, line 27: Please quote the reference to which the reported heights are measured. A1.9 The heights are referred to mean sea level, and this needs to be explicitly quoted.

Q1.10 Page 6, lines 29-31: Unclear. I understand (but I am not sure) that the observed sea level can differ from the forecast represented by the astronomical tide plus the residual sea level ('meteo-marine wave' is bad terminology). The difference does not occur because of local morphology and specific waves, but because the model used for the predictions is not good enough. For instance, it may not include the correct basin bathymetry and coastal morphology, or the atmospheric forcing is too coarse. Anyway, the sentences can be dropped. A1.10 Our intention here is exactly to evidence the limitations of the model. Instead of dropping the sentences, we believe they can be improved, making a specific reference to the coarse resolution of the model in respect to the local morphology.

Q1.11 Page 6, line 32: Both waves and tides are mentioned. It is not clear what 'tides' mean here. A1.11 Effectively here we refer to storm surge, meaning tide+surge.

Q1.12 Page 7, line 6: Do the authors mean Adriatic instead of Italy? Page 8, line 5: Unclear sentence. What are 'the E-R coast values'? A1.12 Another example of sloppy terminology that can be fixed. We refer to the Northern Adriatic coast. With "E-R coast values" we refer to sea level rise at E-R coast.

Q1.13 Page 9, line 14: The IPCC mean sea level rise projections are made for 2081-2100 (central year 2090.5) relative to 1986-2005 (central year 1995.5) (page 7, line 26), that is a 95-year time period, but the authors use a 85-year period. Is that a mistype? A1.13 This is not a mistyping but the consequence of the different starting epochs of the two models, in this case, the reference for subsidence model is almost ten years in advance with respect to the sea level one. To avoid confusion, in the revised manuscript we would explain this difference.

Q1.14 Page 12, line 9: Is 'sea-level component' quoted in comparison to subsidence? Please clarify. A1.14 We should better say "in comparison with subsidence".

Q1.15 Page 14, line 29: Does the model include wind waves set up? These are not mentioned here, while they seem to have been at Page 10, line 16, when they are

distinct from the surge. A1.15 See point Q2.6 below.

Q1.16 Page 15, line 33: The authors should not only say that subsidence rates are assumed unchanged in the 21st century, but also the storminess characteristic. A1.16 We agree; this needs to be specified.

Q1.17 Page 21, Fig. 2: Please say if the zero height in the map corresponds to the 1986- 2005 mean sea level (the IPCC start time), to the zero of the Italian geodetic network or to another thing. I also suggest to use a colour palette that highlights the altitude differences in the low-lying areas. Probably, a 0 m contour could also be useful. A1.17: We agree; Figure 2 can be improved according to the Reviewer's guidelines.

Q1.18 Pages 26-30, Fig. 7-11: The coloured areas are often small compared to the whole figure and most of them are barely visible. Can the authors improve their visibility? A1.18 We agree; Figures 7-11 can be improved following the Reviewer's guidelines.

Q1.19 Page 33, Table 4: In the text (page 10) rare events have a return period Âż100 years, not >100. A1.19 We agree, it should be "»100."

Review 2. G. Le Cozannet

Q2.0 The article by Perini et al. provides an estimation of sea-level rise impacts (in terms of erosion and flooding in the Emilia-Romagna region). Interestingly, the article considers regional subsidence patterns, which have a high spatial variability as shown by previous observations based on SAR interferometry. The article also illustrates how the application of European directives can stimulate studies and discussions regarding the future impacts of climate change. Overall, I think that the article provides an interesting perspective, and that it is relevant to NHESS. What is missing, to my opinion in the article, is a real discussion of the significance of the results and their implications. I can suggest a few recommendations in this respect: A1.0 We thank the Reviewer for his positive response. We acknowledge there is room to improve the manuscript,

especially regarding the implications of the results.

Q2.1 - It could be interesting for the reader to know how such work (which is apparently strongly connected to regulatory processes such as the European flooding directives, e.g. page 3 line 22) will (or is expected to be) integrated in regional to local adaptation. I have no specific suggestion here, but I just remind that the AR6 IPCC reports to come will require information on the implementation of adaptation (including its successes and limitations). I think that the authors can make a useful contribution here. A2.1 We agree that something can be said about this important point. We expect that the output of our analysis will support the planned activities for the second stage of the Flood Directive 2007/60, i.e the updating of the knowledge framework and hazard and risk maps by 2019. These should include, in fact, the risks assessment driven by the climate change, which was not been presented in the first cycle of the directive application, in order to identify mitigation and adaptation measures. Likewise, we expect that this work can provide an important contribution to the working group on the Regional Climate Change Strategy, which aims to develop an action plan by 2018.

Q2.2 - The authors clearly list their assumption all along their study (e.g. section 3.2), but the reader would like to see a discussion on the impacts of these assumptions in the final results. I suggest that uncertainties in the results could be given more attention in a discussion section (see below further suggestions). A2.2 One possible way to answer to this point is i) to express better motivations for the assumptions illustrated in Section 3.2 and ii) to discuss, at least qualitatively, the possible impact of these assumptions in a (new) discussion Section.

Q2.3 - Finally, if possible, it would be interesting to see to which extent this study agrees or disagrees with previous impacts assessments performed in the same region (e.g. Wolf et al., 2016) and why. A2.3 Indeed, a comparison with Wolf et al., 2016 is not straightforward, since goals and approaches of the two works are quite different. However, we agree that something more about this topic can be said in the revised manuscript.

I provide below detailed comments, which are hopefully useful if the authors decide to discuss uncertainties: - Q2.4 Subsidence: I wonder to which extent it is realistic to assume that subsidence is linear in time. In practice, the authors show that it has not been the case in the past (with acceleration in subsidence rates with increased fluid extraction in section 2.1), and this seems to me relatively common in cases of subsidence caused by groundwater extractions (Le Mouelic et al., 2005; Wang et al., 2012; Raucoules et al., 2013). I wonder if the authors would agree that in their table 5, they provide the maximum benefits of an adaptation strategy consisting in mitigating subsidence through reduced fluid extractions. A2.4 Effectively, in the context of this study, the assumption of a linear subsidence can be better motivated. The major variations in rates over time are observed in confined areas, where groundwater exploitation and anthropic impact are particularly strong. In the remaining areas, which are the largest in the study area, the subsidence rates measured in the different monitoring campaigns are similar. We interpret this spatial and temporal distribution as a constant background signal (likely due to natural subsidence) with overlapping interferences due to human activity. These interferences are difficult to model because of the large number of variables and the complexity of the driving processes. In addition, the measures imposed by regional government over the last 30 years to reduce the extraction of fluids from the ground have allowed a progressive mitigation in terms of reducing subsidence rates in most of these critical areas. According to this trend, the latest monitoring can be considered as the worst scenario than possible future ones. These rates were used in modelling as they well describe the geodynamic condition the Emilia-Romagna coast at regional scale, pointing out the present state of the highly subsiding areas.

Q2.5 A small point Page 5 line 12: "compaction of sediments" is unclear to me. I assume the authors refer here to natural (and later, anthropogenic) variability of water content in various geological layers, resulting in a reduction of their volume. A2.5 Yes, we refer to the variability of the water content in the various geological layers.

Q2.6 - Extreme water levels: The authors use value of water heights during storms

(subsection 2.3). However, it is unclear which processes have been incorporated. Of course, the references to the project Micore and other studies suggest that tides, atmospheric surge, wave setup (Stockdon et al. 2006) have been taken into account, but I suggest naming these processes explicitly. Note that the wave setup can account for an additional contribution of several 10cm, which is not negligible considering the magnitude of sea-level changes to come. If no information is available on this process, this source of uncertainties can be assumed dominant for the decades to come. A2.6 The process that have been incorporated are shortly mentioned in Section 3.3.2 (description of case study CS2). They include the wave set up, effectively. We recognise that more information is necessary on this point. We propose to add an equation, in the same section, to help the reader to understand the meaning of SSTS.

Q2.7 - Mean sea-level projections: Sea-level projections used in this article rely on global models, which have not the ability to represent processes taking place at the Gilbratar straight (subsection 2.4). This can result in deviation of some 10 cm from sea-level projections in the Atlantic, west of the Gibraltar straight. Also, is the area affected by 3D circulation modifying water levels by +/-10cm as it is the case in the gulf of Lion? I suggest to discuss these processes in a discussion on uncertainties. They are discussed for example in Adloff et al. (2015, 2016, both in Climate Dynamics) and also in our article Le Cozannet et al. (2015 in Environmental Modeling and Software). A2.7 This limitation of our approach can be addressed in the (new) Discussion section, where we can also account for some of the literature suggested by the Reviewer.

Q2.8 Furthermore, the wording "Worst" or "best" cases scenarios (page 9 line 22 and several times after) is not appropriate for ranges of uncertainties representing likely confidence intervals (see Church et al., 2013a, 2013b) and can be misleading for coastal managers in charge of adaptation (Hinkel et al 2015). This should be rephrased. A2.8 Here and in the following, we propose to use "high-end" and "low-end" instead of "worst" and "best", respectively.

Q2.9 - Impacts : The authors have presented their results in two ways : "land losses"

due to sea-level rise and subsidence (e.g., conclusion) and "areas lying below mean sea-level" (e.g. in table 5). I am personally in favor of the second formulation, as it makes no assumption on the adaptive responses to come (e.g., beach and dunes nourishment. . .). In both cases, the results assume no morphological changes, which, again, would deserve a discussion. There is a huge bibliography in this area. A2.9 The assumption of no morphological changes can be better discussed in the (new) Discussion section. We have chosen to consider a rigid substrate that is only modified by the subsidence (by translation), since, at present, it has not been possible for us to apply morpho-dynamic coastal modeling at the regional scale. In this view, the model excludes the natural adaptation of the coastal system as well as it relies upon the assumption of "no intervention" by man (no nourishment, no upgrading in coastal defense systems, etc.). Of course, this can impact our final results, and result into some uncertainty.

Q2.10 Finally, can the authors explain why storm surge impacts have not been assessed in both CS1 and CS2 hazard assessments (page 14 line 5)? A2.10 With this choice we aim at studying the possible future coastal morphological framework and the effects of the sea storms in this new context separately.

Q2.11 Finally, I wonder if figure 3 and 5 could be merged. A2.11 Probably the best solution is to leave the two figures separated, since Figure (3) contains small scale details that could be obscured by adding the rectangles and the labels in Figure 5. I hope these comments are useful. The suggestions have been useful and we believe that the manuscript could significantly improve following the guidelines suggested by both Reviewers.

References Wolf et al. 2016; Church et al., 2013a: see author's reference list Church et al. 2013b: Church J A, Clark P U, Cazenave A, Gregory J M, Jevrejeva S, Levermann A and Payne A J 2013 Sea-level rise by 2100 Science 342 1445 Wang et al., 2012: Wang, J., W. Gao, S. Y. Xu, and L. Z. Yu (2012b), Evaluation of the combined risk of sea level rise, land subsidence, and storm surges on the coastal areas of Shanghai, China,

Clim. Change, 115(3-4), 537–558, doi:10.1007/s10584-012-0468-7. Raucoules et al., 2013: Raucoules, D., G. Le Cozannet, M. Gravelle, M. de Michele, G. Wöppelmann, M. Gravelle, A. Daag, and M. Marcos (2013), Strong non-linear urban ground motion in Manila (Philippines) from 1993 to 2010 observed by InSAR, Remote Sens. Environ., 139, 386–397, doi:10.1016/j.rse.2013.08.021. Adloff F, Somot S, Sevault F, Jordà G, Aznar R, Déqué M, Herrmann M, Marcos M, Dubois C, Padorno E, Alvarez-Fanjul E, Gomis D; Mediterranean Sea response to climate change in an ensemble of twenty first century scenarios. Climate Dynamics, 2015, Volume 45, Issue 9, pp 2775-280258. Adloff F., Jordà G., Somot S., Sevault F., Arsouze T., Meyssignac B., Li L., Planton S. Improving sea level simulation in Mediterranean Regional climate models. Climate Dynamics. (2016) Le Cozannet G, Rohmer J, Cazenave A, Idier D, van De Wal R, De Winter R and Oliveros C 2015 Evaluating uncertainties of future marine flooding occurrence as sea-level rises Environ. Modell. Softw. 73 44–56 Stockdon, H. F., Holman, R. A., Howd, P. A., & Sallenger, A. H. (2006). Empirical parameterization of setup, swash, and runup. Coastal engineering, 53(7), 573-588. Le Mouelic, S., Raucoules, D., Carnec, C., & King, C. (2005). A least-squares adjustment of multitemporal InSAR data: Application to the ground deformation of Paris. Photogrammetric Engineering and Remote Sensing, 71, 197–204. Hinkel J, Jaeger C, Nicholls R J, Lowe J, Renn O and Peijun S 2015 Sea-level rise scenarios and coastal risk management Nat. Clim. Change 5 188–90.

---

## Author Response (AR1)

Urbino 29 July 2017

Dear Editor,

please find below our response to the Reviewers comments on paper "Sea-level rise along the Emilia-Romagna coast (Northern Italy) at 2100: scenarios and impacts" by Luisa Perini *et al*. (ms No.: nhess-2017-82).

For your convenience, the Reviewers' queries, reported in *italics*, have been numbered as **Qx.y,** where **x** is the Reviewer (**x**=1, 2) and **y** is the point raised. Our response is labeled by **Ax.y**. To facilitate the readability, in the marked manuscript, the answers labels are annotated on the margins.

Most of the references suggested by Reviewer 2 at the end of his letter have been included in the Bibliography in the revised version of the manuscript. Following the suggestions of Reviewer 2, a Discussion section has been added.

Various minor edits have been made throughout the manuscript to improve the presentation; all these are marked in **bold face** but not annotated on the margin. In the marked manuscript, with **[…]** we denote removed text.

We believe that the paper ha greatly benefited from the comments of the two Reviewers.

We hope that this revised version will be suitable for publication, possibly after another round of review.

Sincerely yours

Giorgio Spada  (on behalf of all co-authors)

**Review 1 (Anonymous)**

**Q1.0** *It is an interesting paper that illustrates the consequences of mean sea level rise and storminess on the coastal areas of Emilia-Romagna in terms of land loss. The paper is logically structured and the methodology seems adequate (but I cannot make comments on the flood model because I don't know it). Unfortunately, the paper often contains confusing terminology that makes it hard to understand. This problem should be solved before the paper can be published.*

**A1.0** We thank the Reviewer for his positive response. We acknowledge that the terminology is confusing in some parts of the manuscript, and we have made efforts to improve it.

*Main problems.*

**Q1.1** *a) The authors use 'sea level' both for a long term (multi-decadal) mean and a short term (e.g. hourly) value, as in the case of storm surges. For instance at page 3, line 19, the meaning is 'mean sea level', like at page 9, line 8 (case CS1), where the full 'mean sea level' is used. By contrast, at page 6, line 18, the meaning is 'sea level height' relative to a known reference. As another example, at page 10, line 5, the authors only consider storm surges, while case CS2 also includes the wave effect. Definitions should be clear and used consistently. Note that Sect. 2.2 should involve the 'mean sea level'.*

**A1.1** We recognise that the terminology regarding "sea level" may be inconsistent in some cases: now we have checked the terminology throughout the whole paper.

**Q1.2** *b) The use of 'wave' is also often unclear. For instance, 'storm waves' at page 6, line 19 are clearly wind waves, while 'meteo-marine wave' and 'specific waves' (same page, line 30) are undefined. Please specify when wind waves or any other wave type are involved.*

**A1.2** We thank the reviewer for the remark. Now "waves" is specified correctly, where needed, to avoid ambiguous statements.

**Q1.3** *c) The authors deal with both storm surges and wind waves (case CS2). Sect. 2.3 is devoted to storm surges, but something about wind waves is also included. This is confusing because the role of wind waves is first mentioned explicitly only at page 10, line 16. The authors might consider to deal with storminess in general, that is*

*both storm surges and wind waves (see page 8, line 23), and introduce the subject accordingly.*

**A1.3** We propose to entitle Section 2.3 "Coastal storms", to include both storm surge and waves. The terminology has been changed accordingly throughout the paper. We have now quoted Harley (2017) for the definition of coastal storm.

**Q1.4** *Page 1, line 1: Are there any reasons to neglect natural land subsidence?*

**A1.4** There is no reason to neglect natural land subsidence, effectively. Now we refer to "subsidence", without limiting ourselves to "human-induced".

**Q1.5** *Page 1, line 13: Please clarify that '(in_CoastFlood)' is the name of the model. The missing reference, the different typos and the brackets are confusing.*

**A1.5** This has been fixed in the revised manuscript, in order to clarify that '(in_CoastFlood)' is the name of the model.

**Q1.6** *Page 4, Lines 15-16: 'some cases . . . sometimes' is redundant.*

**A1.6** We agree: "sometimes" has been removed.

**Q1.7** *Pages 5-6, Sect. 2.2: The discussion of mean sea-level changes over thousands of years is not crucial for present and future variations. By contrast, there is no discussion on the mean sea level variations during the last 100 years or so, when also anthropogenic subsidence occurred.*

**A1.7** Effectively, a discussion on the mean sea level variations during the last century was missing. We have added a paragraph on this subject, with reference to recent works about the history of sea level in the Adriatic Sea during last century, due to its relevance with the topics dealt with in the paper. The section "Sea level variability" has benefited significantly from there modifications, in our opinion.

**Q1.8** *Page 6, lines 17, 30: The exact meaning of 'meteo-marine' is unclear in this context. I guess that the authors mean 'the sea level changes component related to the atmospheric forcing', which includes both wind and atmospheric pressure (not*

*mentioned). This component is the 'residual sea level' also known as the 'meteorological tide'. Moreover, is this the 'non-tidal residual' at line 26?*

**A1.8** The entire section 2.3 has been reorganised, and the terms used have been defined in a more thorough way. The term meteo-marine has been avoided and substituted with a more clear terminology.

**Q1.9** *Page 6, line 27: Please quote the reference to which the reported heights are measured.*

**A1.9** The heights are referred to mean sea level, and this has been explicitly quoted in the revised manuscript.

**Q1.10** *Page 6, lines 29-31: Unclear. I understand (but I am not sure) that the observed sea level can differ from the forecast represented by the astronomical tide plus the residual sea level ('meteo-marine wave' is bad terminology). The difference does not occur because of local morphology and specific waves, but because the model used for the predictions is not good enough. For instance, it may not include the correct basin bathymetry and coastal morphology, or the atmospheric forcing is too coarse. Anyway, the sentences can be dropped.*

**A1.10** Our purpose is exactly to enlighten the limitations of the model. Instead of dropping the sentences, we have preferred to improve them, making a specific reference to the coarse resolution of the model in respect to the local morphology.

**Q1.11** *Page 6, line 32: Both waves and tides are mentioned. It is not clear what 'tides' mean here.*

**A1.11** Effectively here we refer to storm surge, meaning tide+surge. See also comment on Q1.3.

**Q1.12** *Page 7, line 6: Do the authors mean Adriatic instead of Italy? Page 8, line 5: Unclear sentence. What are 'the E-R coast values'?*

**A1.12** Our apologies. Another example of sloppy terminology. It has been fixed. We refer to the Northern Adriatic coast. With have also specified that "E-R coast values" refers to sea level rise at E-R coast.

**Q1.13** *Page 9, line 14: The IPCC mean sea level rise projections are made for 2081-2100 (central year 2090.5) relative to 1986-2005 (central year 1995.5) (page 7, line 26), that is a 95-year time period, but the authors use a 85-year period. Is that a mistype?*

**A1.13**  The 85 years time-span is  the consequence of different epoch for the subsidence model with respect to that of the sea level. We understand that this could be misleading and we rephrased the statement omitting this length of the used time-span.

**Q1.14** *Page 12, line 9: Is 'sea-level component' quoted in comparison to subsidence? Please clarify.*

**A1.14** We have now specified "in comparison with subsidence".

**Q1.15** *Page 14, line 29: Does the model include wind waves set up? These are not mentioned here, while they seem to have been at Page 10, line 16, when they are distinct from the surge.*

**A1.15**  See point Q2.6 below.

**Q1.16** *Page 15, line 33: The authors should not only say that subsidence rates are assumed unchanged in the 21st century, but also the storminess characteristic.*

**A1.16** We agree; this has been specified in the revised manuscript.

**Q1.17** *Page 21, Fig. 2: Please say if the zero height in the map corresponds to the 1986- 2005 mean sea level (the IPCC start time), to the zero of the Italian geodetic network or to another thing. I also suggest to use a colour palette that highlights the altitude differences in the low-lying areas. Probably, a 0 m contour could also be useful.*

**A1.17:** The reference height in the map is the zero of the Italian geodetic network. In our opinion the current color table highlights well the elevation differences in low-lying areas.

**Q1.18** *Pages 26-30, Fig. 7-11: The coloured areas are often small compared to the whole figure and most of them are barely visible. Can the authors improve their visibility?*

**A1.18** We agree; in Figures 7-11 the coloured areas are small. We improved the overall quality of figures, from 6 to 10 (new numbering); the current version of manuscript includes these new versions of figures. If necessary, we shall provide higher resolution maps in a later stage of the editing process.

**Q1.19** *Page 33, Table 4: In the text (page 10) rare events have a return period »100 years, not >100.*

**A1.19** We agree, now it is ">>100".

**Review 2. G. Le Cozannet**

**Q2.0** *The article by Perini et al. provides an estimation of sea-level rise impacts (in terms of erosion and flooding in the Emilia-Romagna region). Interestingly, the article considers regional subsidence patterns, which have a high spatial variability as shown by previous observations based on SAR interferometry. The article also illustrates how the application of European directives can stimulate studies and discussions regarding the future impacts of climate change. Overall, I think that the article provides an interesting perspective, and that it is relevant to NHESS. What is missing, to my opinion in the article, is a real discussion of the significance of the results and their implications. I can suggest a few recommendations in this respect:*

**A2.0** We thank the Reviewer for his positive response. We acknowledge there is room to improve the manuscript, especially regarding the implications of the results.

**Q2.1** *- It could be interesting for the reader to know how such work (which is apparently strongly connected to regulatory processes such as the European flooding directives, e.g. page 3 line 22) will (or is expected to be) integrated in regional to local adaptation. I have no specific suggestion here, but I just remind that the AR6 IPCC reports to come will require information on the implementation of adaptation (including its successes and limitations). I think that the authors can make a useful contribution here.*

**A2.1** Something has now been said about this important point. We expect that the output of our analysis will support the planned activities for the second stage of the Flood Directive 2007/60, i.e the updating of the knowledge framework and hazard and risk maps by 2019. These should include, in fact, the risks assessment driven by the climate change, which was not been presented in the first cycle of the directive application, in order to identify mitigation and adaptation measures. Likewise, we expect that this work can provide an important contribution to the working group on the Regional Climate Change Strategy, which aims to develop an action plan by 2018. Now this is clarified in the revised manuscript, see Section Conclusions.

**Q2.2** - *The authors clearly list their assumption all along their study (e.g. section 3.2), but the reader would like to see a discussion on the impacts of these assumptions in the final results. I suggest that uncertainties in the results could be given more attention in a discussion section (see below further suggestions).*

**A2.2** We have decided to discuss, at least qualitatively, the possible impact of the various assumptions made in this work in a (new) "Discussion" Section. This may contribute to a better assessment of the uncertainties implicit in our study, although the establishment of quantitative error bars is out of the scope of present work.

**Q2.3** - *Finally, if possible, it would be interesting to see to which extent this study agrees or disagrees with previous impacts assessments performed in the same region (e.g. Wolf et al., 2016) and why.*

**A2.3** Indeed, a comparison with Wolf et al., 2016 is not straightforward, since goals and approaches of the two works are quite different. However, in the revised manuscript, we have added a specific paragraph on this issue, comparing the two approaches in a somewhat more clearly compared to our previous submission.

*I provide below detailed comments, which are hopefully useful if the authors decide to discuss uncertainties: -*

**Q2.4** *Subsidence: I wonder to which extent it is realistic to assume that subsidence is linear in time. In practice, the authors show that it has not been the case in the past (with acceleration in subsidence rates with increased fluid extraction in section 2.1), and this seems to me relatively common in cases of subsidence caused by groundwater extractions (Le Mouelic et al., 2005; Wang et al., 2012; Raucoules et al., 2013). I wonder if the authors would agree that in their table 5, they provide the*

*maximum benefits of an adaptation strategy consisting in mitigating subsidence through reduced fluid extractions.*

**A2.4** In the revised manuscript, the 1st paragraph of the new Discussion section addressed the issue of the constant rate of subsidence. We now quote the manuscripts suggest by the Reviewer as noticeable examples of human driven subsidence characterised by a variable rate.

**Q2.5** *A small point Page 5 line 12: "compaction of sediments" is unclear to me. I assume the authors refer here to natural (and later, anthropogenic) variability of water content in various geological layers, resulting in a reduction of their volume.*

**A2.5** Yes, we refer to the variability of the water content in the various geological layers.

**Q2.6** *- Extreme water levels: The authors use value of water heights during storms (subsection 2.3). However, it is unclear which processes have been incorporated. Of course, the references to the project Micore and other studies suggest that tides, atmospheric surge, wave setup (Stockdon et al. 2006) have been taken into account, but I suggest naming these processes explicitly. Note that the wave setup can account for an additional contribution of several 10cm, which is not negligible considering the magnitude of sea-level changes to come. If no information is available on this process, this source of uncertainties can be assumed dominant for the decades to come.*

**A2.6** The processes that have been incorporated are shortly mentioned in Section 3.3.2 (description of case study CS2). They include the wave set up, effectively. We have included more information at the end of section 2.3 in the revised manuscript. The meaning of S_STS is well explained in words just after equation (6), and a new equation is probably unnecessary.

**Q2.7** *- Mean sea-level projections: Sea-level projections used in this article rely on global models, which have not the ability to represent processes taking place at the Gilbratar straight (subsection 2.4). This can result in deviation of some 10 cm from sea-level projections in the Atlantic, west of the Gibraltar straight. Also, is the area affected by 3D circulation modifying water levels by +/-10cm as it is the case in the gulf of Lion? I suggest to discuss these processes in a discussion on uncertainties. They are discussed for example in Adloff et al. (2015, 2016, both in Climate*

*Dynamics) and also in our article Le Cozannet et al. (2015 in Environmental Modeling and Software).*

**A2.7** This limitation has been addressed in the (new) Discussion section, where we also account for some of the literature suggested by the Reviewer, which we acknowledge particularly for having done this important point.

**Q2.8** *Furthermore, the wording "Worst" or "best" cases scenarios (page 9 line 22 and several times after) is not appropriate for ranges of uncertainties representing likely confidence intervals (see Church et al., 2013a, 2013b) and can be misleading for coastal managers in charge of adaptation (Hinkel et al 2015). This should be rephrased.*

**A2.8** Here and in the following, we propose to use the wording "HIGH-END" and "LOW-END" instead of "worst" and "best", respectively. The notation for the various equations has been changed accordingly. We hope that this new wording is now suitable, and compatible with Hinkel et al's, otherwise we are looking forward to have new suggestions from the Reviewer. Note that I have now realised that in the insets of Figures 6-10, the previous adjectives "worst" and "best" appear. We shall fix this problem at a later stage of the review process.

**Q2.9** *- Impacts : The authors have presented their results in two ways : "land losses" due to sea-level rise and subsidence (e.g., conclusion) and "areas lying below mean sea-level" (e.g. in table 5). I am personally in favor of the second formulation, as it makes no assumption on the adaptive responses to come (e.g., beach and dunes nourishment. . .). In both cases, the results assume no morphological changes, which, again, would deserve a discussion. There is a huge bibliography in this area.*

**A2.9** Regarding the morphological changes, it was chosen to consider a rigid substrate modified only by the subsidence (translation), since, at present, it has not been possible to apply a dynamic modelling approach integrating also long term coastal evolution (e.g. beach changes) at the regional scale, or land use projection.  In this view, the model excludes the natural adaptation of the coastal system as well as it funds on the assumption of "no intervention" by man (no nourishment, no upgrading in coastal defense systems, etc.). In the discussion Section, these points are addressed. Unfortunately, at the current stage we cannot assess the level of uncertainty due to the assumption made.

**Q2.10** *Finally, can the authors explain why storm surge impacts have not been assessed in both CS1 and CS2 hazard assessments (page 14 line 5)?*

**A2.10** With this choice we aim at studying the possible future coastal morphological framework and the effects of the sea storms separately.

**Q2.11** *Finally, I wonder if figure 3 and 5 could be merged.*

**A2.11** In an effort to improve all the figures of the paper, after several attempts, we have merged (previous) Figure 2 (the DTM) with (previous) Figure 3 (the subsidence map) to obtain (new) Figure 2. The other figures have been improved in minor details (and of course renumbered accordingly) and in the overall resolution so that fine details can hopefully better captured. See also point Q1.18 above.

*I hope these comments are useful.*

The suggestions have been useful and we believe that the manuscript has seen a significant improvement thanks to the help of both Reviewers.

Most of the following references, suggested by Reviewer 2 have been discussed and included into the Bibliography in the revised version of the manuscript.

[revised manuscript text omitted]

---

## Author Response (AR2)

Urbino, Sep. 22 2017.

Dear Editor

please find below our response to the last Referee #2 comments on our paper "Sea-level rise along the Emilia-Romagna coast (Northern Italy) at 2100: scenarios and impacts" by Luisa Perini *et al*. (ms No.: nhess-2017-82). Below, the points are marked by R1, R2, R3, and R4 while our answers are marked by "A".

We have carefully checked the manuscript and found a large number of small errors, sentences that could be improved, typos, etc. These issues have also been addressed, as shown below.

All the modifications are marked in bold face in the annotated manuscript.

We are looking forward to have your feedback.

Your sincerely

Giorgio Spada

Corresponding author for paper nhess-2017-82

**Response to Reviewer 2 and further modifications of the manuscript.**

**Point R1.** Minor comment: the HIGH-END scenario, as defined by the authors page 10, is actually the upper bound of the likely range of IPCC projections. However, high-end scenarios usually refer to sea-level scenarios beyond the likely range (see for example, Hinkel et al 2015 NCC; Nicholls et al 2014 WCC among others). Other documents on this issue could be found in the WCRP Sea level grand challenge documents. I recommend to reconsider this wording all along the manuscript (results, discussion, conclusion, figure 3 in particular).
A: We now use simply "H" and "L" to indicate the two scenarios, to avoid any confusion. We have changed results, discussion, conclusions and other parts of the manuscript accordingly.

**Point R3.** page 2 line 10: high -> highly
A: We have made this substitution.

**Point R4.** page 2 line 15: is "spread" grammatically correct here?
A: Yes, it is correct.

Page 2, lines 14ff. The sentence has been rearranged and simplified.

P2L21. the comma has been moved before "i.e."

P2L29. "The document" cambiamo in "This document".

P3L2. We have placed the callout to footnote outside the quotations.

P3L4. We have changed into "Within the above multidisciplinary path".

**Point R2. M**ay be a word to explain how the terminology on disaster risk reduction and climate change could be useful here (e.g., reference to the ISDR terminology, which uses **mitigation** as a component of the disaster reduction cycle, whereas the word has a completely different meaning in the littérature on climate change
A: To avoid any misunderstanding, we now avoid to write "mitigation" and we have rephrased.

P3L7. An "of" was missing before "coastal".

P3L20. We have added "due to storms" after "flooding".

P3L22. We have slightly rephrased.

P4L32. We have changed "storm surges" into "coastal storms".

P5L2ff.  We have slightly rephrased.

P5L16. We now write "in the region".

P6L12ff. The citations in the paragraph have been displaced to avoid redundancy.

P6L19. We write "presently" instead of "currently".

P6L22 We now write on millennia time scales.

P6L28. H_sig is the correct symbol to be used here, the same as a few lines above.

P6L29: We now write "from east-northwest sectors".

P6L31. We now write "comparatively weak Scirocco".

P7L2. "Emilia Romagna" has been changed into "Emilia-Romagna" throughout the manuscript.

P7L4. A new paragraph is *not* started here, since the text is a natural continuation of the above paragraph.

P8L13. "sea level" is changed into "sea-level".

P8L15. "values is" is changed into "values are".

P8L20. "to year 2100" is changed into "to year 2081-2100".

P8L30. We now use "placement of new artificial barriers" in lieu of "build up", which seems more correct to us.

P9L2. We have slightly rephrased.

P9L5-6 "at along" is changed into "along".

P9L9. "the storminess" is changed into "storminess".

P9L11. "the sea level rise" is changed into "sea level rise".

P9 L11. We write "Emilia-Romagna Regional Administration database" in lieu of "Emilia-Romagna region database".

P9 L18. We now write "This DTM".

P9L20 We have rephrased.

P9L23. A comma has been inserted between "subsidence" and "equal".

P10L1. "case" is omitted.

P10L2. "case" can be omitted.

P10L16. See P7L2.

P10L18. "has been based is changed" into "is based".

P10 L22 We write "run up" instead of "rise", with reference to S_STS.

P10L16. See P7L2.

P11L10. We write "shown" instead "confirmed".

P11L22 "at to" is changed into "to".

P11L22. We now write "impact assessments" instead of "impacts assessments".

P12L16. We have now removed the comma after "2100".

Figure 3. The time label is missing. A refresh to the figure could be useful?

Figures 6 to 10. best-> L, worst-> H.

P13L7. "N" is "northern".

P13L13. We now write "between Cervia and the  Fiumi Uniti river mouth".

P13L16. We change "the sea level rise" with "sea level rise".

P13L22. We have deleted the short sentence "Furthermore, in the case of flooding…" since the absence of defences is already mentioned at L23 in the same page.

P13L17-18. The previous sentence was a bit cumbersome. We have rephrased "However, the storm surge would flood the entire coast also in this area. The frontal ingression could extend several hundred meters, exceeding 0.5 km in the P2-DTM12 scenario, thus involving large urban areas".

P13L22. We have changed "below the mean sea level" into "below mean sea level".

P14L3. We have changed "from land" into "of land".

P14L9. "defenses" is "defences".

P14L15. We now better say "of enlarged floodable areas".

P14L19. This is actually "more than two times"?.

P14L22. Item five (v) is now listed using the same style as for the others.

P15L2. We have removed ":" after "regarding".

P15L3. "required" is "requires".

P15L13. We write "of the critical areas" instead of "of these critical areas".

P15L17.  We have rephrased to makes the text more fluent.

P15L18 "Regarding" is substituted by "Concerning".

P15L21. "it funds on" is replaced by "it is based on".

P15L22. "in coastal" is replaced by "of the coastal defence".

P15L26. "is uncertain" is replaced by "are uncertain".

P16L1: The comma is deleted after "in particular".

P16L3. "of file" is changed into "life".

P16L17. "effects of effects" is "effects".

P16L29ff. We have rephrased with awkward sentence.

P27, Caption of Figure 6. A dot was missing at the end.

Bibliography: Some typos have been corrected.

Table 1. It has been slightly edited.

[revised manuscript text omitted]